


# The usefulness of Extended-Range Probabilistic Forecasts for Heat wave forecasts in Europe

Natalia Korhonen[1], Otto Hyvärinen[1], Virpi Kollanus[2], Timo Lanki[2,3,4], Juha Jokisalo[5], Risto Kosonen[5,6], David S. Richardson[7], and Kirsti Jylhä[1]

[1] Weather and Climate Change Impact Research, Finnish Meteorological Institute, Helsinki, Finland
[2] Unit of Environmental Health, Department of Health Security, Finnish Institute for Health and Welfare, Kuopio, Finland.
[3] School of Medicine, University of Eastern Finland, Kuopio, Finland.
[4] Department of Environmental and Biological Sciences, University of Eastern, Kuopio, Finland.
[5] Department of Mechanical Engineering, Aalto University, Espoo, Finland
[6] College of Urban Construction, Nanjing Tech University, Nanjing, China
[7] European Centre for Medium-Range Weather Forecasts (ECMWF), Reading, UK

*Correspondence to*: Natalia Korhonen (Natalia.Korhonen@fmi.fi)

**Abstract.**

Severe heat waves lasting for weeks and expanding over hundreds of kilometres in horizontal scale have many harmful impacts on health, ecosystems, societies, and economy. Under the ongoing climate change heat waves are becoming even longer and hotter, and as proactive adaptation, the development of early warning services is essential.

Weather forecasts in extended range (2 weeks to 1 month) tend to indicate a higher skill in predicting warm extremes than average temperature events in Europe. We verified hindcasts of the European Centre for Medium-Range Weather Forecasts (ECMWF) in forecasting heat wave days, i.e., periods with the 5-day mean temperature being above its $90^{th}$ percentile. The verification was done in $5° \times 2°$ resolution over Europe, based on the forecast week (1 to 4 weeks). In the first forecast week, it is evident that across Europe, the accuracy of ECMWF heat wave forecasts surpasses that of a mere climatological forecast. Even into the second week, in many places in Europe, the ECMWF forecasts prove to be more reliable than their statistical counterparts. However, if we extend the forecast lead time to 3-4 weeks, predictability begins to lower to such a level that it can no longer be said, with the exception of Southeastern Europe, that the forecasts in general were statistically significantly better than the statistical forecast. Nonetheless, intense and prolonged heat waves during the third forecast weeks appear to have a higher-than-average level of predictability.

## 1 Introduction

The severest heat waves in Europe since the 1950s have lasted from several weeks to even longer than a month, with horizontal spatial ranges exceeding several hundred kilometres, even 1000 km (Russo et al. 2015). In recent decades the number of extreme heat waves over Europe and across the Northern Hemisphere has increased and in future, due to the ongoing climate





change, heat waves are expected to become even more common and intense (IPCC, 2021; Russo et al. 2014; Coumou and Rahmstorf, 2012; Kim et al. 2018, Vogel et al. 2020, Ruosteenoja and Jylhä, 2023).


Prolonged heat waves have negative impacts on, e.g., human health and wellbeing (Arsad et al. 2022; Guo et al. 2017), particularly so in urban areas (Ruuhela et al. 2021; Gasparrini et al. 2022, Kivimäki et al. 2023); labour productivity (Kjellstrom et al. 2009; Dunne et al. 2013; Orlov et al. 2019); energy and water resources (Añel et al. 2017; Hatvani-Kovacs et al. 2016; van Vliet 2023); transport systems (Mulholland & Feyen, 2021), wildfire safety (Rossiello and Szema, 2019; Ruffault et al.,
2020);  agriculture (Heino et al. 2023; Vogel et al. 2019), and livestock (Ahmed et al. 2022, ; Morignat et al. 2014). Prolonged and intensive heat waves occurring over a wide area can lead to significant, and potentially catastrophic, impacts on public health. In Europe, the 2003 heat wave has been estimated to have resulted in over 70 000 (Robine et al. 2008) and the 2022 heat wave in over 60 000 (Ballester et al. 2023) heat-related deaths. As climate change progresses, severe health effects of heat waves are expected to further increase (Guo et al. 2018).


There is no commonly shared definition of a heat wave. However, in epidemiological studies examining heat-related health effects, heat waves are usually defined as periods when the daily temperature exceeds the 90[th] or higher percentile of the local annual or summertime temperature distribution for two or more consecutive days (Arsad et al. 2022). These types of heat waves have been observed to lead to increased mortality and morbidity all around the world (Arsad et al. 2022, Guo et al.
2017), also in countries with cool climatic conditions (Kollanus et al. 2021, Sohail et al. 2020). Heat stress and dehydration can lead to mild or severe heat illnesses and exacerbate symptoms of chronic illnesses (Ebi et al. 2021). Vulnerable population groups include the elderly, chronically ill people, infants and small children, pregnant women, and those working in hot environments.

Heat waves are particularly harmful to health when heat exposure lasts for several days or weeks. During heat waves, apartments lacking air conditioning gradually begin to overheat, which exacerbates heat stress (Velashjerdi Farahani et al. 2021; Velashjerdi Farahani et al. 2023; Velashjerdi Farahani et al. 2024a). In Northern Europe apartments are typically not equipped with mechanical cooling systems.  In a Finnish study, heating of buildings has been observed to take 5-6 days, during which the thermal mass of the building was able to slow down the indoor temperature rise (Velashjerdi Farahani 2024a). The
urban microenvironment, including factors such as the green view index, floor area ratio (the ratio of a building's total floor area to the total area of the land upon which it is built), and distance from the sea, influence indoor temperatures in buildings to some extent especially during short heat waves (Kravchenko et al. 2023).

While it is crucial to mitigate the ongoing climate change and alleviate the tendency towards more frequent and intense heat
waves, it is also increasingly vital to focus on reduction of risks posed by these (Curtis et al. 2017; Martinez et al. 2019; Rapeli and Mussalo-Rauhamaa 2022; Matzarakis and Nouri 2022). During the past 20 years, many countries in Europe and other



parts of the world have taken steps to improve heat preparedness by developing heat-health action plans, which aim to reduce the negative health impacts by implementing measures to protect vulnerable population groups and the general population (Kotharkar et al. 2022; Martinez et al. 2022). A key element of these preparedness plans consists of heat wave early warning

systems, the operation of which is based on weather forecasts and pre-defined threshold criteria for triggering the warning services (Casanueva et al., 2019; Prodhomme, et al. 2021). Effectiveness of the systems in preventing health effects depends on the ability to accurately forecast the impending heat event, as well as warning lead time, which determines how much time the respondents have available to prepare and take the required actions. The lead time for heat wave warnings in each European country depends on the respective National Meteorological and Hydrological Services. Currently, heat wave warnings across

Europe are typically issued 2-5 days in advance, and in some countries, such as Germany and the U.K., up to 7 days in advance.

The skill of the extended range forecasts (2 weeks to 1 month, also called sub-seasonal forecasts) has been found to be atmospheric flow dependent (Frame et al. 2013, Ferranti et al. 2015) and spatially heterogeneous. Vitart & Robertson (2018) highlighted the potential of sub-seasonal predictions in forecasting the progression of prolonged events like heat waves

spanning multiple weeks. Moreover, Wulff and Domeisen (2019) and studies by Pyrina and Domeisen (2023) emphasized that extended-range predictions were more successful in forecasting extreme hot summer temperatures in Europe compared to predicting average summer temperatures. Our objective was to assess the accuracy of forecasts made by the European Centre for Medium-Range Weather Forecasts (ECMWF) in predicting *heat wave days*, defined as periods where the local 5-day mean temperature exceeded the 90[th] percentile of the local summertime 5-day mean temperature distribution. We assessed the

reliability of forecasts predicting heat waves surpassing this threshold, as this type of heat waves have been shown to significantly increase the risk of overheating in apartments in Finland (Velashjerdi Farahani et al. 2024a) and elevate mortality risk among the elderly (Kollanus et al. 2021). Moreover, in an empirical study conducted in Finland, indoor temperatures were found to be more strongly correlated with outdoor 5-day moving average temperature than with average temperatures of a few days only, suggesting impacts of building's thermal inertia (Velashjerdi Farahani et al., 2024b). Our verification process was

conducted using a resolution of 5 degrees longitude and 2 degrees latitude ($5° \times 2°$) over Europe for the summers spanning from 2000 to 2019. We examined forecasts for various lead times, ranging from 1 to 4 weeks. The novelty of the study arises from the verification area encompassing the entirety of the European region, the verification time span spanning 20 years of summers, and the use of a 5-degree by 2-degree resolution, which allows for the highlighting of potential regional differences in forecast skill. In Section 2 we define *heat wave days* and the metrics used to evaluate them. In Section 3 we show the model

skill and in Section 4 we discuss them and in Section 5 we summarize our results.





## 2 Materials and Methods

### 2.1 Hindcasts and verification data

Hindcasts, also known as reforecasts, are a type of retrospective weather forecasts. Hindcasts are forecasts of past weather conditions, generated using forecasting models, data assimilation methods, and observational data identical to those used for real-time weather predictions. By comparing the hindcasts to actual historical weather data, the skill of the forecasting system can be evaluated. We verified summer (June, July, August) extended range hindcasts of the European Centre for Medium-Range Weather Forecasts (ECMWF) Integrated Forecasting System (IFS; Cycles 46r1 and 47r1; Vitart, 2014). These hindcasts were run at the ECMWF in 2020 twice a week, on Mondays and Thursdays. We investigated 240 hindcasts which were run with a weekly interval for the summers 2000–2019, i.e., 20 years × 12 weeks =240 hindcasts. The hindcasts consisted of a control forecast and 10 perturbed ensemble members. We examined the 2m temperature (i.e. the near-surface air temperature) from the hindcasts with lead times of 1 to 32 days of the Monday runs. Since the hindcasts were initiated using the ERA5 analyses, we used the ERA5 2000-2019 near-surface air temperature reanalyses (Hersbach et al., 2020) for verification. Here the area of interest was Europe (36 to 70° N and -7.5 to 52.5° E). The original horizontal resolution of the ECMWF's hindcasts was 0.4° and of the ERA5 reanalyses 0.1°, and here these data were bilinearly interpolated to resolution 5° × 2° including only land grid points.

### 2.2 Definition of heat wave days

Utilizing the ECMWF's 240 hindcasts, we calculated the 5-day moving average temperatures, $T_{EC}^{5d}$. This was done separately for each (5° × 2°) grid point. The calculations of $T_{EC}^{5d}$ were performed separately for each of the 11 ensemble members, covering each day from June 1st to August 27th (88 days) over the summers of 2000-2019. For each day within this period, we incorporated forecasted mean temperatures for that day and the subsequent four days into the calculations of $T_{EC}^{5d}$. For each grid point and for each forecast week, ranging from week 1 to week 4, we determined the threshold for *a heat wave day* by calculating the 90th percentile, the $^{90th}T_{EC}^{5d}$, of the 5-day moving average temperatures, $T_{EC}^{5d}$, of the summers 2000-2019. The forecast data used for the forecast weeks were partially overlapping due to the use of 5-days moving averages with forward-looking window: the forecast week 1 used data of days 1 to 11, the forecast week 2 data of days 8 to 18, forecast week 3 data of days 15 to 25, and forecast week 4 data of days 22 to 32.

The forecasted probability of a heat wave day, *p*, was based on fitting a normal distribution to the $T_{EC}^{5d}$ forecasts of the 11-member ensemble and defining the probability of the forecasted $T_{EC}^{5d}$ being above the $^{90th}T_{EC}^{5d}$ on each day. The use of percentiles for the hindcasts leads to forecasting in the model's climatology. Moreover, the comparison of the hindcasts to the lead time dependent model climatology is expected to remove the systematic bias resulting from the forecast model drift (Manzanas, 2020). It is important to distinguish between the hindcasts, consisting of 11 members, and the operational real-time forecasts, which initially had 51 members and now consist of 101 members (IFS Cycle 48r1). When computing





probabilities using raw ensemble members, this could significantly influence the results (see, e.g., Richardson 2001, Ferro et al. 2008). However, since we fit a normal distribution to the 11-member ensemble, the impact of ensemble size on the results is likely reduced. Nevertheless, the small ensemble size is likely to have some effect, especially on the spread, i.e. the standard deviation of the normal distribution. The results obtained here from the 11-member hindcasts thus serve as a baseline measure of skill. The operational ensemble (51 members in the version under study, and 101 members in the current version at the time of writing) is expected to provide improved estimates of the normal distribution parameters, thereby enhancing skill to some extent.

For verification of these forecasts, we separately defined the 5-day moving average temperature in the ERA5 ($T^{5d}_{ERA5}$) data in the corresponding grid points over Europe, in summers 2000-2019, and defined periods with $T^{5d}_{ERA5}$ exceeding its 90th percentile ($^{90th}T^{5d}_{ERA5}$) as observed heat wave days. To identify the summer with the longest heat wave, we examined the frequency and duration of heat wave days in the ERA5 reanalysis data. A heat wave was considered to be any period of at least one day where the 5-day moving average temperature remained above the 90th percentile of $T^{5d}_{ERA5}$. The heat wave was considered interrupted when there were two consecutive days with temperatures falling below the 90th percentile of $T^{5d}_{ERA5}$. To clarify, a single day below the threshold did not end the heat wave as long as it continued afterward.

In epidemiological studies on heat-related health effects, heat waves are typically defined as periods when *daily temperatures* exceed the 90th percentile of the local annual or summertime temperature distribution for *two or more consecutive days* (Arsad et al. 2022). Such heat waves have been observed to lead to increased mortality and morbidity all around the world (Arsad et al. 2022, Guo et al. 2017). As our definition for the heat waves was based on 5-day mean temperatures, we examined the proportion of days where the daily mean temperature, the $T^{1d}_{ERA5}$, exceeded its 90th percentile ($^{90th}T^{1d}_{ERA5}$) within periods with $T^{5d}_{ERA5}$ exceeding its 90th percentile here defined as heat waves.

**2.3 Skill scores**

The Brier Scores (*BS*, Brier, 1950) of the probabilistic forecasts, *p*, were calculated separately for each grid point and for forecast weeks 1 to 4 as follows:

$$BS = \frac{1}{N}\sum_{t=1}^{N}(p_t - o_t)^2 ,$$ (1)

where $p_t$ is the forecasted probability of a heat wave day, *p*, ranging from 0 to 1, $o_t$ is the actual outcome (based on ERA5 reanalysis) of the heat wave day at instance *t* (0 if there is no heat wave day and 1 if there is a heat wave day), and *N* is the number of forecasting instances. The *BS* is thus here equivalent to the mean squared error of the probability (of the heat wave day), and ranges from 0 to 1. The lower the *BS*, the better the predictions.





It follows from the definition used here for a heat wave day that its expected value, or climatological base rate, is 0.1. This was
  used as the reference forecast when calculating Brier Skill Score (*BSS*):

$$BSS = 1 - \frac{BS}{BS_{ref}} , \qquad (2)$$

where $BS_{ref}$ is the *BS* of the reference forecast calculated by Eq. (1). The value of the *BSS* ranges from -∞ to +1: positive values
indicate better skill than that of the reference forecasts, and *BSS* value of 1 represents the best possible score.


Initially, we calculated the Brier Skill Score (BSS) for each grid point using data from all 240 hindcasts. For a more detailed
evaluation of forecast accuracy during prolonged heat wave periods, we then specifically computed the BSS for each grid point
using data solely from the summer characterized by the longest continuous heat wave. We define this period as the longest
heat wave (as detailed in Section 2.2). Importantly, this analysis was conducted separately for each grid point, acknowledging

that the summer with the longest heat wave may vary from one grid point to another. To demonstrate the impact of the summer
with the longest heat wave on the overall *BSS* of all hindcasts, we also determined the *BSS* while excluding the data from this
specific summer.

For each grid point and lead time, we determined whether the hindcasts were considered more skilful than the reference

forecasts by assessing the *BSS* using a bootstrap resampling procedure. The *BSS* was required to be statistically significantly
above zero for the hindcasts to be considered more skilful than the reference forecasts. To assess statistical significance, we
calculated the p-value under the null hypothesis that the *BSS* is zero. This calculation was performed using bootstrap resampling
with replacement and a sample size of 5000. Because the statistical test on the map is repeated many times, small p-values are
bound to occur by change alone and the null hypothesis is rejected too often. Unadjusted p-values therefore overestimate the

results (Wilks, 2016). We adjust the p-values using the false discovery rate (FDR) method. Technically, the FDR controls for
the expected proportion of false discoveries (hypotheses that should not have been rejected) among the rejected hypotheses.
By setting the threshold to 0.1 (twice the conventional 0.05, as suggested by Wilks 2016), and using the Benjamini-Hochberg
procedure (e.g., Benjamini and Hochberg, 1995), we ensure that on average no more than 10% of the rejected null hypotheses
are false discoveries.

**2.4 Verification by probability ranges**

Weather forecasts can be divided into two main categories: deterministic and probability forecasts. Deterministic forecasts
provide a single specific scenario for future weather. For example, "tomorrow will be hot" is a deterministic forecast that offers
one possible future event. Probability forecasts, on the other hand, provide various possible scenarios and their probabilities,
taking into account the uncertainty of the forecast. For instance, "50% chance of heat" is a probability forecast indicating that

heat may occur, but it's not certain.





In this study we examined the realization of heat wave days across all grid points in Europe when the forecasted probability of heat wave days fell within the ranges of 0 and 0.33, 0.33 and 0.66, and 0.66 and 1.

**2.5 Predicting the lifecycle of a heat wave**

To investigate how early in the forecasts a heat wave days become discernible, we categorized all of the ERA5 dataset days at each grid point based on the phase of the heat wave or whether the day fell outside of it. If the day was a heat wave day, we examined which day of the ongoing heat wave it was at that grid point. If the day was not a heat wave day, we examined how close (temporally) it was to the temporally nearest heat wave day at the same grid point during that summer, classifying it as

1...over 21 days before/after the heat wave. If there were no heat wave days during the entire summer at that grid point, the temporal distance to the nearest heat wave day during all the heatless days of that summer were classified as "over 21 days before the heat wave."

Subsequently, we analyzed the probabilistic heat wave day forecasts ($p$) to determine the probabilities the forecasts provided:

when the timeframe preceded a heat wave (1...over 21 days), when the timeframe was within a heat wave (1st day of a heat wave...over 5th week of heat), or when the timeframe was after a heat wave (1...over 21 days).

**3 Results**

**3.1 Thresholds for the heat wave days**

Figure 1 depicts maps of the 90th percentile of the 5-day moving average temperature (in summers 2000-2019) over Europe,

in ERA5 (Fig. 1a) and in the ECMWF hindcasts for forecast weeks 1-4 (Figs.1b-1e). Days having ERA5 5-day moving average temperatures above the thresholds, the 90th percentile, were in this study defined as heat wave days. The ECMWF hindcasts capture the northwest-southeast gradient in the threshold of the heat wave days, even though the absolute values are somewhat lower in the hindcasts than in ERA5, and this difference is growing with the lead time.



**The 90th percentile of the 5 days moving average temperature (°C) in summers 2000-2019**

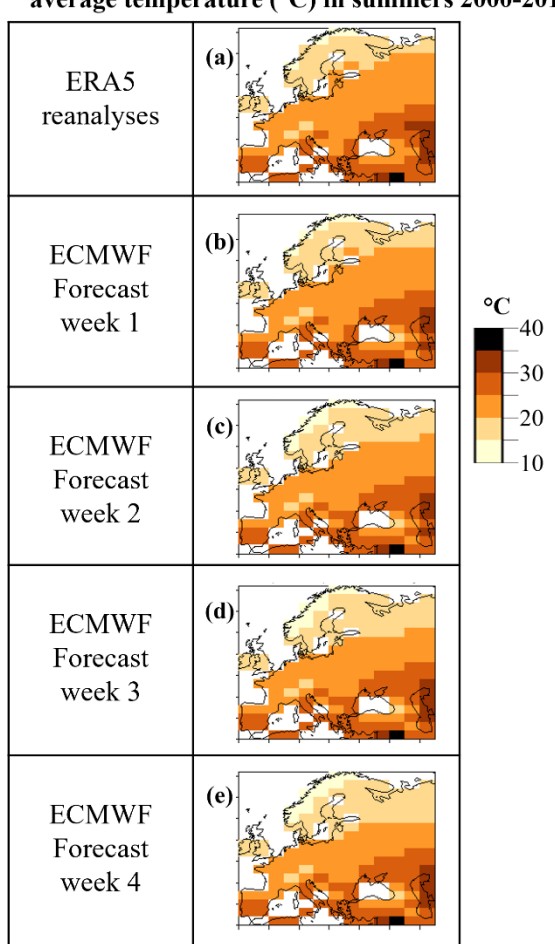

**Figure 1: The lower thresholds of heat wave days: the 90th percentile of the 5-day moving average temperature in summers 2000-2019 (a) of the ERA5 reanalyses, and (b) to (e) of the ensembles of the ECMWF's hindcasts in different forecast weeks.**

To investigate the effect of using 5-day mean temperatures in defining heat waves in comparison to using daily mean temperatures, we computed also daily mean temperatures, the $T_{ERA5}^{1d}$, and their 90th percentiles ($^{90th}T_{ERA5}^{1d}$) across European land areas from 2000 to 2019. For each grid point, we determined the percentage of five-day periods exceeding the $^{90th}T_{ERA5}^{5d}$ that included days where the $T_{ERA5}^{1d}$ exceeded its 90th percentile. The results in Table 1 show that our definition for heat waves based on exceeding the $^{90th}T_{ERA5}^{5d}$, covered 26% of the one-day heat waves based on exceeding the $^{90th}T_{ERA5}^{1d}$. For the two-days heat waves based on exceeding the $^{90th}T_{ERA5}^{1d}$, our definition covered 61%. The three-days heat waves based on exceeding the $^{90th}T_{ERA5}^{1d}$ were covered 96% by our definition. For four or more consecutive day heat wave events based on exceeding the $^{90th}T_{ERA5}^{1d}$, our definition covered 100%.





**Table 1. The proportion (%) of heat waves of different lengths, based on exceeding the $^{90th}T^{1d}_{ERA5}$, included among the heat waves based on exceeding the $^{90th}T^{5d}_{ERA5}$.**

| The length of the heat wave (days) based on exceeding the $^{90th}T^{1d}_{ERA5}$ | The proportion (%) of these heat waves based on exceeding the $^{90th}T^{1d}_{ERA5}$ among the heat waves based on exceeding the $^{90th}T^{5d}_{ERA5}$ |
|---|---|
| **1 or more (All)** | **88%** |
| 1 | 26% |
| 2 | 61% |
| 3 | 96% |
| 4 | 100% |
| 5 or more | 100% |


## 3.2 The frequency and duration of the heat wave days

The durations of the longest heat wave events in each grid point over Europe in summers 2000-2019, as derived from ERA5, are depicted in Fig. 2(a). The heat wave events were longest in Eastern Europe. Figure 2(a) highlights the extreme heat wave of 2010 in the east, the heat wave of 2018 in the north and parts of Central Europe, and the heat wave of 2003 in parts of south

and southwest. Figure 2(b), showing the number of different heat wave events, highlights that in these summers 2000-2019 the heat wave days in Northern Europe and in many parts of Eastern Europe were concentrated within fewer periods, whereas in the Central and Southwestern Europe, the same amount of heat wave days were distributed across a larger number of periods.





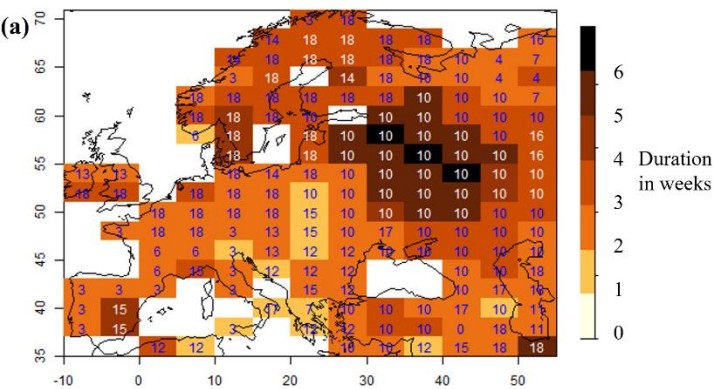

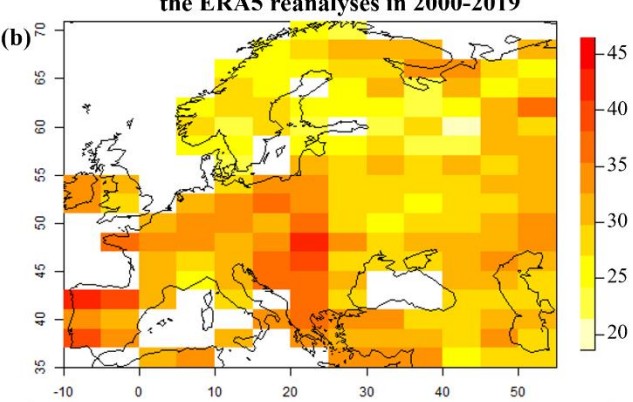

**Figure 2: The duration and the year of the longest period of heat wave days (a) in each grid point over Europe defined from the**
**ERA5 reanalysis data of summers 2000-2019 (marked as 0-19), and the number of periods with heat wave days (b) in the ERA5**
**reanalyses during 2000-2019.**

**3.3 Reliability of probabilistic heat wave days forecasts**

The sharpness diagram in Figure 3(a) displays the relative frequency with which the heat wave days were predicted (in the
hindcasts of summers 2000-2019 in all land grid points over Europe) with different levels of probability (0.1, 0.2, …, 0.9, 1).
If all the forecasts were perfect, then in Fig 3(a) 90% of the forecasts would have $p=0$ and 10% would have $p=1$. For the first
week, in Fig 3(a) roughly 80% of the forecasts belong to the lowest probability class and 5% to the highest one. However, as
the lead time increases, both these portions decrease, while the share of forecasts with $0.1<p<0.9$ increases. The sharpness of
forecasts drops as the lead time increases. Regarding the forecasts for heat wave days (as depicted in Figure 3a), it is worth
noting that while approximately 5% of the forecasts for the one-week lead time have $p>0.9$, the relative occurrence of such
forecasts is nearly negligible for weeks 2 to 4.




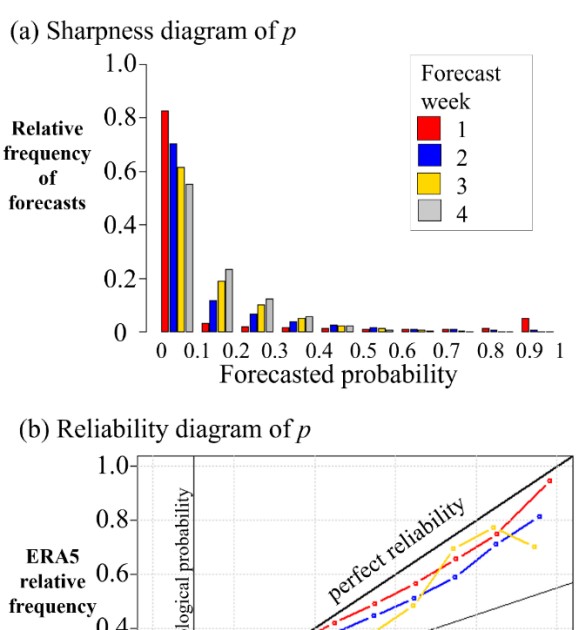

**Figure 3: (a) sharpness diagram and (b) reliability diagram of the 1-4 weeks probabilistic heat wave days forecasts, *p*, over Europe (all land grid points) in summers 2000-2019.**

In Fig. 3(b), we illustrate the relationship between forecasted probabilities and observed frequency of heat wave days by aggregating data from all grid points to create a reliability diagram. The forecasted probabilities are displayed on the x-axis and observed frequencies on the y-axis. In a perfectly calibrated forecast, the points on the reliability diagram would fall along a 45-degree diagonal line from the bottom left to the upper right corner. This line represents *perfect reliability*, where the forecasted probabilities match the observed frequencies. The *climatological probability* line in the reliability diagram

represents the expected frequency of a heat wave days (0.1) based on climatology. The *no skill* line in the reliability diagram represents the baseline for a forecast with no skill. In other words, it represents what a forecast would look like if it had no value beyond random chance or climatology. The points on the reliability diagram above the *perfect reliability* line indicate underforecasting, meaning that the forecasted probabilities are too low compared to the observed frequency. Conversely, the points on the reliability diagram below the perfect reliability line indicate overforecasting, meaning that the forecasted

probabilities are too high compared to the observed frequency.





The reliability of the heat wave day forecasts was best for shorter forecast weeks and dropped by growing lead times (Figure 3(b)). During forecast weeks 1 and 2, the overall reliability of heat wave day forecasts across Europe (as illustrated in Fig. 3b) was nearly flawless when $p<0.4$. Subsequently, for $p>0.4$, the forecasted probabilities tended to be slightly elevated compared to the observed frequencies, suggesting a tendency toward overforecasting. Nevertheless, the reliability remained higher than that achieved by climatology alone. For forecast week 3 the overall reliability of the heat wave day forecasts over Europe (Fig. 3d) was nearly the same as that of climatology for $p<0.5$, thereafter for $p>0.5$ the forecasted probabilities showed improving reliability in comparison to the reference forecast ($p=0.1$), however indicating overforecasting.

### 3.4 Probabilistic prediction skill scores for heat wave days

Figure 3b depicted the average reliability of the heat wave days predictions over the whole of Europe. Next, we will take a look at the forecast skill across different regions over Europe to find out how the accuracy varies in different regions. First, we assess the performance of the 240 hindcasts of all summers from 2000 to 2019. Second, we focus on summers characterized by the longest heat wave (12 hindcasts/summer), and third, we examine hindcasts of summers from 2000 to 2019 excluding the hindcasts of the summer with the longest heat wave (228 hindcasts). In the first column of Figure 4, we present the *BSS* of all hindcasts of the summers 2000-2019. During the first forecast week, the predictions of heat wave days in Europe demonstrates strong performance, with *BSS* values ranging between 0.5 and 0.8. These values are statistically significantly superior to the reference forecast at every grid point. However, in later forecast weeks, the skill diminishes. In the second forecast week, BSS ranges from 0.1 to 0.4 in Europe, remaining statistically significantly better than the reference forecast in most grid points across the continent. The exceptions include certain grid points over the northern parts of the Iberian Peninsula, eastern central Europe, and northeast of the Caspian Sea. Moving to forecast weeks 3 and 4, BSS values in Europe range between -0.1 and 0.2, exhibiting statistical significance only in specific grid points across Eastern and South-Eastern Europe.

In the middle column in Fig. 4, we have depicted the *BSS* for each grid point of only those hindcasts of the summer with the longest heat wave, as defined from the ERA5 data and depicted in Fig. 2. In the forecast weeks 1 and 2 the *BSS*s of the hindcasts of the summers with the longest heat waves were higher than the *BSS*s of the hindcasts of all summers 2000-2019 especially in the Mediterranean, Eastern Europe, and Northern Europe where in the forecast week 1 the *BSS* are 0.1 to 0.4 higher than the *BSS* for all summers, and in the forecast week 2 the *BSS* was about 0.1 to 0.6 higher. Moving to forecast weeks 3 and 4, the *BSS* of hindcasts of the summer with the longest heat wave exceeded the *BSS* for all summers 2000-2019 by 0 to 0.5 in Northern Europe (summer 2018), Eastern Europe (summer 2018), and in parts of the Mediterranean coast (summer 2015), even reaching values of 0.5.

In the last column in Fig 4, we illustrate the BSS for each grid point of all hindcasts excluding the summer with the longest heat wave days. The BSS excluding the heat wave summer differs mostly only +/-0.05 from the BSS of all summers, except in Eastern Europe where the BSS is even 0.1 lower in forecast weeks 2-4. In more detail: in the first forecast week, the BSSs




of the hindcasts excluding the summer with the longest heat wave are between 0.4 and 0.7 and in all grid points statistically significantly higher than 0, i.e., better than the reference forecast. In the second week the BSS of the hindcasts excluding the summer with the longest period of heat wave days are between 0 and 0.4 and still statistically significantly higher than 0 in the majority of the grid points. In the third and the fourth week, however, the BSS is statistically significantly higher than 0 only in some grid points in southeastern parts of the map.

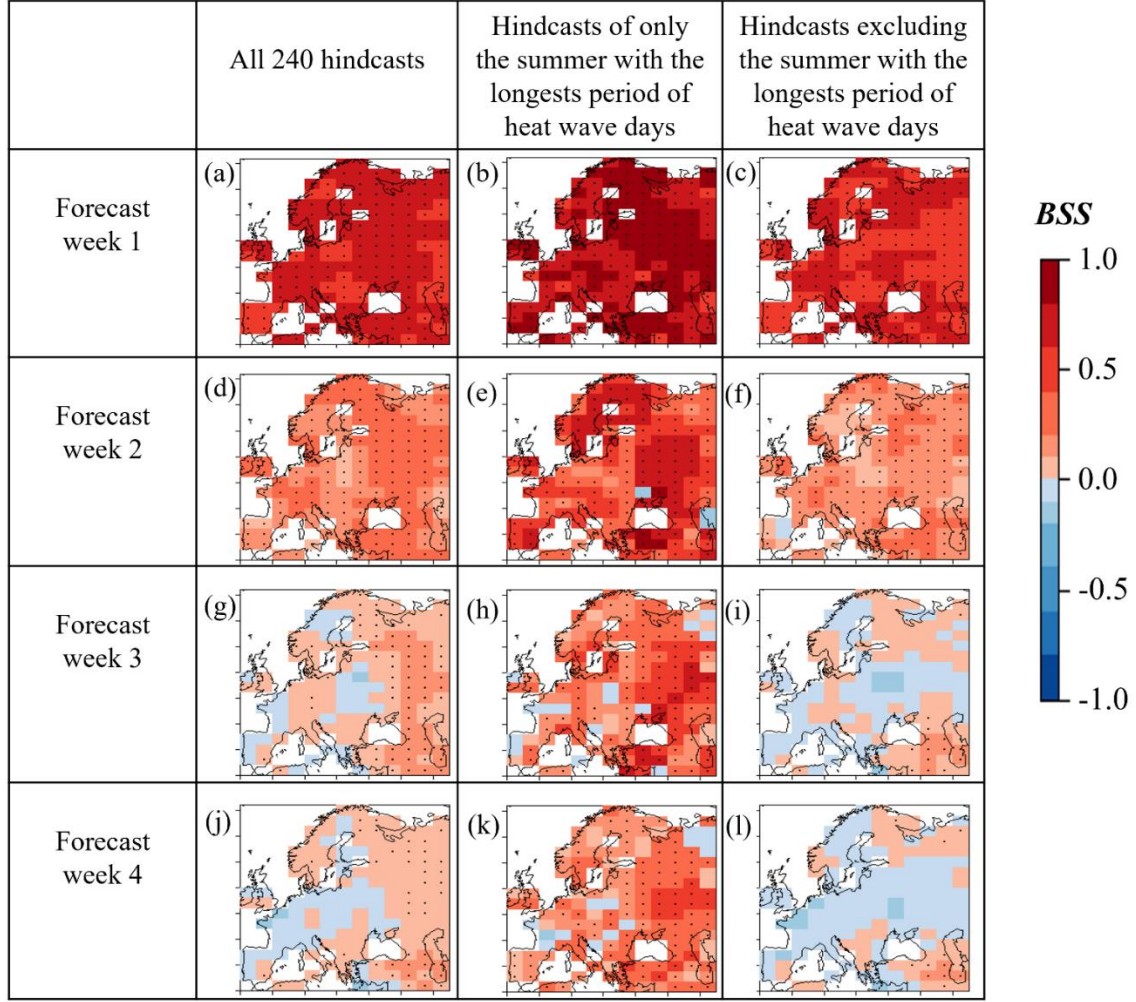


**Figure 4: Brier Skill Scores (*BSS*) of the probabilistic heat wave days forecasts, *p*, during all summers 2000-2019 (first column), during the summer with the longest period of heat wave days of summers 2000-2019 defined from the ERA5 reanalysis data (middle column), and in hindcasts excluding the summer with the longest period of heat wave days. The statistical occurrence *p*=0.1 for heat wave days were used as the reference forecasts. The dotted areas show where *BSS* is greater than zero with the false discovery rate 310 no more than 10%.**





### 3.5 Verification by probability ranges

We conducted verification of heat wave day forecasts across all grid points in Europe based on forecasted probabilities falling within the ranges of 0 - 0.33, 0.33 - 0.66, and 0.66 - 1. In Figure 5, boxplots depict the observed ERA5 temperature (percentiles)
in each forecasted probability range. The parts of the boxplot above the 90th percentile (grey horizontal line) indicate heat wave days in the ERA5 temperature reanalysis. It is important to note that each boxplot has a different amount of data marked as *n* above each box. The category with the most forecasts is within the 0 and 0.33 range.

When forecasting with a one-week lead time, we observed that heat wave days occurred in 2%, 45%, and 86% of cases,
corresponding to forecasted probabilities falling within the ranges of 0-0.33, 0.33-0.66, and 0.66-1, respectively. Moreover, when forecasting with a two-week lead time, heat wave days occurred in 7%, 39%, and 68% of cases, corresponding to forecasted probabilities falling within the ranges of 0-0.33, 0.33-0.66, and 0.66-1, respectively. When forecasting with a three-week lead time, heat wave days occurred in 10%, 30%, and 67% of cases, corresponding to forecasted probabilities falling within the ranges of 0-0.33, 0.33-0.66, and 0.66-1, respectively. When forecasting with a four-week lead time, heat wave days
occurred in 11%, 28%, and 38% of cases, corresponding to forecasted probabilities falling within the ranges of 0-0.33, 0.33-0.66, and 0.66-1, respectively. Hence, higher probabilities (p>0.66) show that a heat wave event is more likely, however for forecast weeks 3 and 4 the forecasting signal is not very strong due to the relatively low proportion of *n* (amount of data) in group p>0.66.

**Figure 5: Boxplots of the ERA5 5-day moving average temperature over Europe in each grid point across different levels of $p$ (the forecasted probability of a heat wave day) with lead times of a) one week, b) two weeks, c) three weeks, and d) four weeks. The horizontal line dividing each box into two parts shows the median of the data; the ends of the box show the lower and upper quartiles; and the whiskers indicate the 5th and 95th percentiles of the ERA5 data in each group. The width of each box and the $n$ written above each box indicate the number of observations in each group. The notches of each side of the boxes were calculated by R boxplot.stats. If the notches of two plots do not overlap, then this is strong evidence that the two medians differ (Chambers et al., 1983). The grey horizontal line indicates the 90th percentile, i.e., the threshold of a heat wave day, and the percentiles above (and below) the grey line depict the amount of observed heat wave days (and no-heat wave days) after the different levels of forecasted probability.**





### 3.6 Predicting the lifecycle of a heat wave

In Figure 6, the forecasted probabilities of heat wave days are shown for days categorized as explained in Section 2.5: "before the heat wave," "during the heat wave," and "after the heat wave", across the entire European region at each land-grid point. Dashed green boxes delineate forecasts where, at the time of issuance, a heat wave in that grid point is about to begin within a week. Solid green boxes indicate forecasts where, at the time of issuance, a heat wave is ongoing in that grid point. If the forecasts perfectly aligned with reality, the probability forecast should be zero in the categories "before the heat wave" and

"after the heat wave," and in the category "during the heat wave," the probability forecast should be 1 (i.e., 100%).

In heat wave day forecasts four weeks in advance (lead time 22...28 days) (Figure 6d, Forecast week 4), there are only slightly higher probabilities for the heat wave during the heat wave than before and after. Particularly, a small portion of the data where the fifth week of the heat wave (days 29..35) is in progress, shows higher probabilities. These forecasts are in the "green box,"

indicating ongoing heat when the forecast was issued.

In heat wave day forecasts three weeks in advance (Figure 6c, Forecast week 3), higher probabilities for days within the heat wave are already more apparent than for days outside the heat wave. Especially for the third, fourth, and fifth weeks of heat wave days, higher probabilities are evident compared to non-heat wave days. These forecasts are in the green box area,

indicating that the heat wave is just starting or already underway when the forecast is issued.

In heat wave day forecasts two weeks in advance (Figure 6b, Forecast week 2), forecasts again show more clearly higher probabilities for heat wave days for days within the heat wave than outside. Especially as the heat wave extends to the second, third, fourth, fifth, or more weeks, forecasts show significantly higher probabilities. Additionally, there is some slight

overestimation for heat wave days in the days just before and after the heat wave, indicating slight inaccuracy in forecasting the exact day of the start and ending of the heat wave.

In heat wave day forecasts one week in advance (Figure 6a, Forecast week 1), considerably higher probabilities for heat wave days for all lengths of heat waves and throughout the heat wave are evident compared to non-heat wave days. Additionally,

there is some overestimation, particularly 1-2 days before or after the heat waves indicating slight inaccuracy in forecasting the exact day of the start and ending of the heat wave.


**Figure 6: The forecasted probabilities of heat wave days shown for days that (in ERA5) were 21 to 1 days before the heat wave, the 1st to 35th heat wave day during the heat wave, and 1 to 21 days after the heat wave with lead times of a) one week, b) two weeks, c) three weeks, and d) four weeks. Dashed green boxes indicate forecasts where, at the time of issuance, a heat wave in that grid point was about to begin within a week. Solid green boxes indicate forecasts where, at the time of issuance, a heat wave was already ongoing in that grid point. The boxplots (as in Fig.5) include all forecast data across the European region at each land-grid point.**

## 4 Discussion

We examined the skill of hindcasts of the ECMWF in forecasting heat wave days over Europe 1 to 4 weeks ahead. The verification was done against ERA5 reanalysis. Heat wave days were here defined as days with the 5-day moving average temperature being above its summer 90th percentile. The hindcasts covered 20 summers of the period 2000-2019. The verification was done in $5° \times 2°$ resolution. The assessed hindcasts demonstrated varying levels of accuracy across different forecasting lead times and regions, which is in line with many earlier studies, e.g. Ferranti et al. (2015), Wulff and Domeisen (2019), and Pyrina and Domeisen (2023).




The overall performance of the hindcasts in predicting heat wave days of summers 2000-2019 over Europe can be summarized as follows:

- Forecast week 1, i.e., 1 to 7 days in advance: strong performance in predicting heat wave days.
- Forecast week 2, i.e., 8 to 14 in advance: still statistically significantly better than the reference forecast in most
grid points over Europe.
- Forecast weeks 3-4, i.e., 15-28 days in advance: statistically significantly better than the reference forecast only in specific grid points across Eastern and South-Eastern Europe.

Across all forecast lead times, ranging from 1 to 4 weeks, the skill scores (BSS) for hindcasts of summers with the longest heat waves surpassed those for all summers from 2000 to 2019 by up to 0.5. Noteworthy enhancements were particularly observed

in Northern Europe, Eastern Europe, and specific regions along the Mediterranean coast (Fig. 4).

The heat wave days forecasts seem to have high potential in warning of heat risk in 1-2 weeks in advance. Additionally, the probabilistic heat wave day forecasts show enhanced accuracy in forecasting prolonged (here several weeks long) heat waves, in lead times of up to three weeks at the time that the heat wave had initiated prior to the forecast issuance (Fig. 6). Further, as

in all forecast weeks 1-3 there was a higher-than-average predictability for intense and prolonged heat waves in Northern Europe, Eastern Europe, and specific regions along the Mediterranean coast (Fig. 4).

Verification by probability range (in Figure 5) shows that at least for lead times for 1-2 weeks, there is signal that lower probability (probabilities below 0.33) forecasts could be valuable for indicating periods when it is unlikely that a heat wave

will occur. And the higher probability (probabilities above 0.66) forecasts could be valuable for indicating periods when a heat wave could occur.

Heat waves cause severe health risks and have considerable impacts on public health, particularly if the high temperatures last for many days or weeks and occur over a wide area (Ballester et al. 2023, Robine et al. 2008). The health impacts can be

reduced by timely short-term measures to protect the vulnerable population groups (Martinez et al. 2019, Tooloo et al. 2013). The needed measures are preferably outlined in national or sub-national heath-health actions plans, as recommended by the World Health Organization (WHO) (Martinez et al. 2019, Matthies et al. 2008). Currently existing plans include many individual and community-level actions, such as communication of warnings and public health recommendations; safeguarding homes and facilities from overheating; preparing health and social care services and emergency operations and increasing their

capacity; establishing public cooling centres and shaded outdoor spaces; identifying and contacting vulnerable people who need support or special arrangements; ensuring the safety precautions of public and sporting events; setting up a telephone helpline; and occupational safety measures (Casanueva et al. 2019; de'Donato et al. 2018; Kotharkar et al. 2022; Lowe et al. 2011, Martinez et al. 2022).





Heat-health action plans rely on early-warning systems to inform all stakeholders and citizens on the impending heat wave and to activate the public health interventions in time. As health effects of heat exposure occur quickly, at the same day or a few days lag (Baccini et al. 2008), it is imperative that the protection measures are implemented rapidly when a potentially dangerous heat wave is forecasted. However, organisation of the response measures requires coordination of actions between many stakeholders and distribution of workforce, equipment, and other resources, which take time. Currently, most heat

warning systems in Europe have lead-times of only a few days (Casanueva et al. 2019). Hence, the response operations would benefit from the 1–2 weeks, and in some cases possibly even longer, heat wave days forecasts.

Prolonged hot periods may pose concurrent risks that challenge functioning of society, including health care system and other critical infrastructure. The negative impacts on energy and water supply, transportation, and agriculture and livestock can

affect human health and wellbeing more indirectly and on a wider scale. Prolonged heat waves and drought also lead to wildfires (Parente et al. 2018), which can be catastrophic events leading to immediate losses of life and high economic costs (San-Miguel-Ayanz et al. 2013). Moreover, severe air pollution episodes from the fires may have significant effects on population health over long distances (Kollanus et al. 2017). In the event of a far reaching and catastrophic heat wave, management of all negative impacts simultaneously would require large-scale interagency and cross-border response, as well

as significant mobilisation of human and other resources. Heat wave days forecasts of 1-2 weeks lead time have potential to support preparedness for and coordination of these types of civil protection efforts in Europe.

As climate change is projected to increase the number of hot days across Europe (e.g., Ruosteenoja and Jylhä, 2023), society, including Northern Europe where many homes and public buildings lack cooling systems, should prepare for rising need for

preventing overheating. To mitigate the risk of exacerbating climate change through increased electricity consumption on a large scale, it is advisable to prioritize passive methods in preventing overheating whenever they are sufficient. In urban areas, wise urban design such as green areas, preventing excessive building density, and bodies of water can mitigate the urban heat island effect (Kravchenko et al. 2023). Let it be noted that increasing green areas can also serve to reduce urban flood risk (Zimmermann et al. 2016). In buildings, priority should be given to passive cooling methods such as sunshades, proper and

timely ventilation, and possibly the color of the building's exterior. Many of the passive cooling methods can be retrofitted into existing buildings. Furthermore, when passive methods are insufficient to prevent overheating indoors, active cooling is to be considered (e.g., Velashjerdi Farahani et al. 2021).

In this study, to address potential systematic bias resulting from forecast model drift, we opted to use percentiles separately

for ERA5 and the ECMWF's hindcasts. Although this approach was anticipated to mitigate bias, further improvements in forecast skill could be achieved through additional bias correction techniques. For example, utilizing so-called windows of opportunity (e.g., Büeler et al 2021, Mariotti et al. 2020, Vigaud et al. 2019) or other sources of predictability for European





heat waves as introduced to result from weather regimes (Büeler et al., 2021), the North Atlantic Sea surface temperatures (Cassou et al. 2005; Duchez et al. 2016), El Niño–Southern Oscillation (Wulff et al. 2017), the North Atlantic Oscillation
(Kenyon and Hegerl 2008), the Atlantic Multi-decadal Oscillation (Della-Marta et al. 2007), propagation of Rossby waves (Fragkoulidis et al. 2018; Wolf et al. 2020), the land-atmosphere interaction (Lorenz et al. 2013; Miralles et al. 2014; Ardilouze et al. 2019), and the Boreal Summer Intraseasonal Oscillation (Rouges et al. 2023) could offer avenues for refinement.

**5 Conclusions**

Our examination of ECMWF hindcasts for predicting heat wave days (periods where the local 5-day mean temperature
exceeded the 90th percentile of the local summertime 5-day mean temperature distribution) of summers 2000-2019 across Europe, 1 to 4 weeks in advance, showed varying accuracy levels across forecast lead times and regions, aligning with previous research. The examined ECMWF's hindcasts showed:

- in the first forecasts week (1 to 7 days in advance): strong forecast skill in predicting heat wave days,
- in the second forecast week (8 to 14 days in advance): statistically significantly better skill than the reference forecast
in most grid points over Europe,
- in the forecast weeks 3-4: statistically significantly better skill than the reference forecast only in some grid points across Eastern and South-Eastern Europe,
- in all forecast weeks: a higher-than-average predictability for intense and prolonged heat waves in Northern Europe, Eastern Europe, and specific regions along the Mediterranean coast, and
- in the forecast weeks 1-3: enhanced accuracy in forecasting prolonged (here several weeks long) heat waves at the time that the heat wave had initiated prior to the forecast issuance.

These findings underscore the potential of these ECMWF's heat wave days forecasts to serve as early warnings for impending heat risks 1-2 weeks in advance. Notably, the higher-than-average predictability for intense and prolonged heat waves (at the
time they have already started), offers a potential to early warnings even at a 3-week lead time. However, it is crucial to highlight the known uncertainty in the 3-week lead time forecast.

Early warnings are important in anticipating heat risks. However, mitigating the negative impacts of the anticipated heat waves also requires tailored early action plans and financial support to implement them. As an overall increase in hot days across
Europe due to climate change is anticipated, there is a need for society, particularly in regions lacking cooling infrastructure, to adopt both active and passive approaches for preventing overheating. The prioritization of passive methods, such as green spaces and sensible urban planning, are recommended to mitigate the risk in exacerbating climate change through increased electricity consumption, complemented by retrofittable passive cooling techniques for buildings.



**Data availability**

The ERF data of the ECMWF's IFS cycles 46r1 and 47r1 were retrieved from the ECMWF's MARS archive at https://apps.ecmwf.int/mars-catalogue/ (MARS, 2024). The ERA5 reanalysis data were retrieved from the Copernicus Climate Change Service Climate Data Store (CDS) at https://cds.climate.copernicus.eu/cdsapp#!/home (CDS, 2024).: date of access 11th April 2024. The data of Figs. 1–6 are available at https://fmi.b2share.csc.fi/records/8ceb3da6ce144180b5304ad080af88af (Korhonen, 2024).

**Author contribution**

NK led the writing of the paper and produced figures 1-6. All authors contributed to writing the manuscript.

**Competing interests**

The authors declare that they have no conflict of interest.

**Acknowledgements**

We acknowledge the ECMWF for the forecast data and all those involved in producing ERA5 data. We acknowledge Matti Kämäräinen from the Finnish Meteorological Institute for valuable comments during this research.

**Financial support**

This study is part of the following projects: HEATCLIM (Heat and health in the changing climate, Grant Numbers 329304, 329305, 329306, and 329307) within the CLIHE (Climate change and health) program, and ACCC (Atmosphere and Climate

Competence Center, Flagship Grant No. 337552) both funded by the Research Council of Finland.

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
