# Peer review of "The Probabilistic Skill of Extended-Range Heat Wave Forecasts Over Europe"

_Natural Hazards and Earth System Sciences, 2024_

## Referee Comment (RC2)

Review nhess-2024-75 "The usefulness of Extended-Range Probabilistic Forecasts for Heat wave forecasts in Europe"

This study investigates the probabilistic skill of extended-range forecasts of mildly extreme land temperatures over Europe. It shows that these forecasts are overall reliable out to the third forecast week, but, except for Eastern/Southeastern Europe, do not significantly differ in skill from a much simpler climatological forecast. The skill of the forecasts appears to be strongly enhanced by the most long-lasting events. Excluding these events results in reduced skill over almost all of Europe. An analysis of the evolution of skill throughout the life cycle of the heat wave indicates that the models capture the persistence of anomalous temperatures well, whereas the onset and end of the events seem more difficult to predict.

The study presents a relevant contribution to the field of evaluation of extended-range/subseasonal prediction for potentially impactful events. While previous studies have considered the prediction skill of the same extended-range forecasts for extreme temperatures before, this study adds a thorough assessment of the *probabilistic* skill of the forecasts by using some well-documented methods and scores (which facilitates comparability) and providing some more non-standard ways of looking at the prediction skill. Assessing the probabilistic instead of the deterministic skill of the forecasts is arguably much more important in the extended range, since their uncertainty is large, but the information in the spread of the ensemble could still make the forecasts reliable. The employed methods are sensible and the skill analysis for the heat wave life cycle is innovative and a highlight of the paper. I do, however, not entirely agree with the way the study is framed. The title implies that the study assesses the usefulness of the forecasts, which I don't think it does. The authors also stress the health impacts of heat waves a lot, which is of course a good motivation to investigate the skill at predicting heat extremes, but the study does not include any analysis that links the forecasts to health impacts/heat stress in any way. Furthermore, some of the methods should be explained and motivated more clearly. Finally, the writing could be made more concise in many places. I provide more detailed comments below.

Major remarks:

1. The word "usefulness" in the title made me as a reader expect something (more related to climate services) that is not shown in the study. The usefulness of a forecast can only be determined by involving its user(s), which also means that the forecast, in most cases, will be useful only to some but not others. Furthermore, for a forecast to be useful, skill (which is what I think the article is actually focused on) is just one of many requirements. So, unless the analysis is extended significantly and involves this component, I suggest changing the word "usefulness" to something else here (maybe "skill" would be most accurate).

2. There is a lot of text concerning health impacts/risks of heat in the discussion (ll. 403 – 443). While I don't generally disagree with anything that is written about this, I don't think it deserves the amount of space it is given in the discussion, given there is no direct relationship with the presented results. The study

investigates the probabilistic skill of summer forecasts for mildly extreme (dry bulb) temperatures and the discussion should focus on this aspect. The authors offer an explanation for why they use the temperature measures that they use, and I think it is fair to focus on these, but there is evidence indicating that other measures of temperature are more strongly related to heat stress (involving radiation, humidity, wind) and thus more suitable for measuring health risk/impact of heat events, see e.g. Di Napoli et al. (2019), McGregor & Vanos (2018). Thus, I would suggest removing the too detailed discussion of health impacts of heat from Section 4. Alternatively, if the focus on health impacts should be kept, I suggest considering the use of other, possibly more heat-stress-related, metrics.

Di Napoli, C., F. Pappenberger, and H. L. Cloke, 2019: Verification of Heat Stress Thresholds for a Health-Based Heat-Wave Definition. *J. Appl. Meteor. Climatol.*, **58**, 1177–1194, https://doi.org/10.1175/JAMC-D-18-0246.1.

McGregor, Glenn R., and Jennifer K. Vanos, 2018: Heat: a primer for public health researchers, *Public Health*, **161**, 138-146, https://doi.org/10.1016/j.puhe.2017.11.005

3. If the current focus of the paper is kept, I think the discussion needs to be revised strongly. As mentioned above, the part ll. 403 – 443 seems very detached from the results of the study right now. The remaining text in Section 4 (ll. 374 – 401) is more of a summary and is to a large degree repeated in Section 5 (where it belongs, in my opinion). I think this part could be used better to discuss the implications, the potential and the limitations of your study (as you do in ll. 444 - 452), see below for some suggestions:

   - One question that I wonder about when seeing the results (although it is beyond the scope of the paper to answer this finally): Could it be that the forecasts are generally too persistent and thus lucky when a long-lasting heat wave happens, or do they actually "know" when to persist temperatures? In other words, are they right for the right reasons? The fact that the exclusion of the most long-lasting events basically removes all remaining skill from the week 3 & 4 forecasts makes me think that they might just have been lucky. Also, your Figure 6 could be interpreted further with this question in mind.
   - You mention climate change in the discussion (l. 433). Against the backdrop of climate change, what do your results mean? Are we expecting better forecasts because we will see more (and potentially longer-lasting) heat extremes? Or might the predictability of these events also change?
   - Parts of your manuscript suggest that you would like to link this to the applicability of extended-range forecasts in early warnings of heat waves (e.g. l. 391). Could you elaborate on what your results mean, e.g. for an agency that would want to implement these forecasts for early warnings? Is the skill sufficient? Can the presented aggregation over large geographical areas (5˚ x 2˚) be useful in some way? Where can the

forecasts contribute and where can't they, keeping in mind that they are ok at predicting the persistence but not so good at predicting the onset far in advance?

4. Since you try to address usefulness/applicability of the forecasts, it could be a good idea to assess reliability on a regional ("grid-point") level in addition to the *BSS* (Fig. 4). The reason is that reliability can be linked better to decision-making, see Weisheimer & Palmer (2014). Their paper shows a simple method of categorizing forecasts by the slope (and its uncertainty) of their reliability curve into 5 categories. This would address the usefulness aspect at least to some degree and could be a nice addition to the current results.

   Weisheimer, A., and T. N. Palmer, 2014: On the reliability of seasonal climate forecasts, *J. R. Soc. Interface*, **11**: 20131162. http://dx.doi.org/10.1098/rsif.2013.1162

5. In general, Section 3 could use some additional explanations to make the results of the analysis easier to grasp for the reader. Generally, at the beginning of each subsection (3.X.), provide one sentence on why we're seeing this plot now and what it's supposed to tell us (like you do in ll. 276 – 277). More specifically:

   i. Section 3.5: Since this is not a very standard form of presenting forecast skill (at least not one I'm familiar with), I suggest explaining the reason for showing the skill in this form. I get the feeling it is relatively closely related to the reliability diagram. In what way does it differ/provide extra information? What can we learn from this way of looking at the forecasts? As a reference for the reader, give an example of what a good and a poor forecast would look like if displayed in this way (as you do in the part with the reliability diagram). A bit more information on this could also aid the interpretation of the next plot.

   ii. Figure 6, Section 3.6: I consider the life cycle plot a highlight of the manuscript, but it contains a lot of information, so I think it deserves a more thorough discussion (and to be picked up in Section 4!). One thing I find particularly noteworthy in this figure is that, while there seems to be an upward trend in the forecast probabilities leading up and into the heat waves, the highest probability class ($p > 0.66$) is only really predicted when the heat wave is already present in the initialization of the forecast.

Minor comments:

*Title*

"forecast" is used twice, could maybe reformulate?

*Intro*

l. 26: 'intense and prolonged heat waves during the third forecast weeks' The study doesn't really address intensity, so the first part of this should be removed. I also think it would be more accurate to say that persistence of heat/extreme temperatures seem to have a higher level of predictability. The current sentence suggests that the forecasts are generally (onset, duration, intensity, ending) better for strong events.

l. 28: one sentence linking back the results of the study to the motivation (early warning systems) would round off the introduction a bit more.

l. 32: 'in future' to 'in the future'

l. 37: 'particularly so in urban areas' can be removed since there is no relation of this to the question the study addresses.

ll. 46 – 54: I think it should be mentioned here that high (dry bulb) temperature is only one factor in heat stress, see references I provide above.

ll. 55 – 63: I understand this paragraph as a motivation to consider the prediction of longer-term averages of temperature. If that's the case, be more explicit about it and say that due to the above reasons there could be value in considering the prediction of these averages. This could also be related to the fact that longer aggregations might be better predictable, see e.g.

> Toth, Z. and R. Buizza (2019). "Weather Forecasting: What Sets the Forecast Skill Horizon?" In: *Sub-Seasonal to Seasonal Prediction: The Gap Between Weather and Climate Forecasting*. Ed. by A. Robertson and F. Vitart. 1st. Elsevier. Chap. Chapter 2, 17 –45.

ll. 59 – 62: I don't see the relevance of this with regards to the study. Can be removed.

ll. 64 – 75: This fits more into the general motivation of the study at the beginning of the intro (potentially in a shortened form)

l. 64: 'alleviate the tendency towards more frequent and intense heat waves' I don't understand what this means.

ll. 82 – 85: work out more clearly what your study is adding and providing beyond what has been done previously. Stress the probabilistic nature of the forecasts that you are evaluating and the analysis of the 'heat wave skill life cycle'

ll. 86 – 89: This is already mentioned in ll. 55 – 62 and does not need to be repeated here

l. 91: change 'forecasts' to 'hindcasts' or 're-forecasts'

l. 94/95: These two sentences seem a bit redundant as they are now. Can you be a bit more specific in guiding the reader through the paper here?

*Methods*

l. 96: The word 'Materials' seems a bit off in the context of the study. Maybe 'Data' is more appropriate?

ll. 100 – 101: This could maybe be formulated more carefully. The skill of the hindcasts gives an indication of the skill of the forecasting system, but it is not necessarily the same (as you point out in ll. 126 – 134, so maybe merge these sentences).

ll. 101: Meaning all forecasts initialized during JJA (which includes forecast and verification for September days) or all with verification dates in JJA?

ll. 102 – 104 & l. 106: What is the reason for only using Monday initializations instead of all available ones?

l. 109: The ECMWF (re-)forecasts are run at higher horizontal resolution up to day 15 and then re-initialized at lower resolution from day 15 to 46.

ll. 112 - : I suggest starting with defining heat wave days for the verification since the verification data is simpler (it only has one time dimension). Then you only have to explain how you handle the extra time dimension (lead time) in the hindcasts.

l. 117: Is this the 90$^{th}$ percentile of all (summer) days under consideration or for each calendar day individually?

l. 125: bias → frequency bias

ll.127 – 134: Maybe this could be re-structured a bit because it seems to be going back and forth between saying the hindcast ensemble is large enough to get an idea of the forecasting system's skill and saying it is not.

l. 134: Another important difference between the skill shown in the study and the skill of the actual forecasting system is that in forecast mode, there is no information about the future, while you are using all years (including the evaluated one) when defining the percentiles. This is likely to lead to an overestimation of the skill. To simulate this setting, a leave-one-year-out cross validation could be employed. I'm not requesting the authors to do this, but I think it should be pointed out in addition.

ll. 141 – 142: This sentence sounds like it is stating the obvious. Maybe better to say something like: "A single below-threshold day between two heat wave days was nevertheless classified as a heat wave day."

ll. 144 – 149: see comment on Table 1 below.

l. 168: do you mean "define this period as *the summer containing* the longest heat wave"? Is the entire summer taken out or just the period of the longest heat wave?

ll. 175 - 1178: Could you provide a more detailed description of how the bootstrap resampling procedure works?

l. 179: "change" → "chance"

ll. 182 – 183: Explain in a few words how this procedure works.

ll. 184 – 190: This seems to be better placed in the part where you explain how you generate a probabilistic forecast from the ensemble.

ll. 191 – 192: Why these categories? They seem rather arbitrary. Are they used somewhere, which would justify considering them here?

l. 196: "a heat wave days become discernible" I don't understand this, please reformulate

ll. 203 – 205: This part is a bit difficult to understand (especially before having seen Figure 6). Maybe reformulate this.

*Results*

ll. 210 – 211: I think the information in this sentence is redundant here and already given where it is relevant.

ll. 219 – 226, Table 1: What do you conclude from these numbers and how is this relevant for the forecasts or even their skill? Maybe this could rather become part of the method section (2.2.) if the point is to justify the definition of heat waves using the 5-day mean. To me, it wasn't clear why I'm seeing the table at this point in the paper. Since the information in the table is also entirely contained in the text, you could consider removing the table.

ll. 231 – 237: The same as the above comment applies to this subsection. This is just looking at ERA5, so it has nothing to do with the forecasts. I suggest moving this to Section 2 where the heat wave definition or the exclusion of the longest events is described. Alternatively, dedicate a short section at the beginning of Section 3 to the analysis that only deals with ERA5.

l. 245: I think its noteworthy that this is not valid the other way around. You aren't claiming that, but I think it helps a reader who might be less familiar with the details of forecast verification to stress that sharpness is a property of the forecasts alone, i.e. 90% forecasts with $p = 0$ and 10% with $p = 1$ does not directly imply a perfect forecast (i.e. sharpness is a necessary but not a sufficient condition).

l. 259: match → equal

l. 267: by → with

l. 268: can drop the parentheses, it is mentioned in the sentence before.

ll. 270 – 271: "reliability remained higher than that achieved by climatology alone" → this statement cannot be true since by the way you define climatology (i.e. without leaving the validation year out) it has perfect reliability by definition (but no resolution).

ll. 271 – 273: I think there is a mix-up here between the "no skill-line" and the reliability of climatology. Climatology (as defined here) has perfect reliability, so no forecast can possibly have better reliability. It does, however, not have any resolution (it predicts $p = 0.1$ in all instances) and so its $BS$ is higher than 0. If points lie above the "no skill-line" it means that they contribute positively to the $BSS$ with climatology as reference. This is comparing the $BS$ of the forecast to the $BS$ of climatology, not just the reliability. For details see:

> Mason, S. J., 2004: On Using "Climatology" as a Reference Strategy in the Brier and Ranked Probability Skill Scores. *Mon. Wea. Rev.*, **132**, 1891–1895, https://doi.org/10.1175/1520-0493(2004)132<1891:OUCAAR>2.0.CO;2.

l. 280/281: "the predictions [...] demonstrates" → "the forecasts [...] demonstrate"

l. 282: superior to the reference forecast → different from 0

l. 284: as before, here you basically say "BSS remains better than the reference forecast" while what you mean is that the BSS remains above zero, or alternatively, the forecasts remain better than the reference.

Figure 4, ll. 288 – 295: While I think excluding the summers with the longest heat waves gives a good idea of how strongly the overall skill of the forecasts is influenced by these events, I don't think we can learn much from the skill for just the summer with the longest heat wave. While it seems to be in line with the conclusions from the right column in Fig. 4, I would argue that all the middle column might be telling us is that the reference forecast is particularly bad when you choose to basically look at one event alone (meaning $o_t$ in the $BS$ is 1 most of the time and thus the $BS$ of climatology, i.e. $p_t = 0.1$, gets very high, because now your climatological forecast is not reliable anymore). Unless of course you recalculate the 90th percentile using only one summer, which is obviously problematic (representativeness), too.

l. 317: refer back to Figure 3a?

Figure 5: Why is the total $n$ (sum of $n$ for all 3 categories) for each subplot different? Shouldn't this add up to the total number of forecast days within each forecast week times the number of considered grid points?

Ll. 334 – 335: I don't quite understand what is meant by the notches here. The second sentence rather belongs into the results with a description of where we see this in the plot and what it implies.

Section 3.6: I find it a bit confusing that the results are described from the longest to the shortest lead time here, when throughout the rest of the paper, the description starts with week 1. Maybe an option to invert the order?

l. 349: no need to put the "green box" in quotation marks.

ll. 368 – 372 (caption Figure 6): what are the limits of the box plots? Same as in Figure 5, i.e. interquartile range and whiskers for 5$^{th}$ and 95$^{th}$ percentile?

l. 448: "as introduced to result from"; I don't understand what this means.

l. 451: "the land-atmosphere interaction" → "land-atmosphere interactions"

ll. 444 – 452: Could you be more specific about how this could be used to refine the forecasts?

ll. 458 – 462: This is almost an exact repetition of ll. 383 – 387. Keep it only in one place (I'd suggest Section 5).

ll. 473 – 478: Like the aforementioned part of the discussion (ll. 403 – 443), this paragraph seems very detached from the core results of the paper. Rather end the conclusions with some outlook for future work and how it could be continued to make it even more relevant in the context you bring up here.

---

## Author Comment (AC1)

**MS. Ref. No.: NHESS-2024-75 "The usefulness of Extended-Range Probabilistic Forecasts for Heat wave forecasts in Europe"**
**Natural Hazards and Earth System Sciences**

We appreciate your constructive comments as they help to enhance the quality of our manuscript. The following are point-by-point answers in blue colour:

**Referee #1 (RC1):**

**General comments:**
This study focuses on the sub-seasonal predictability of heat waves over Europe using the ECMWF model. The variable studied here is 'heat wave days,' which refers to the number of 5-day periods whose average temperature exceeds the 90th percentile of the climatological distribution. This approach highlights certain performances beyond week 2, particularly for the most intense and prolonged episodes, and mostly over the eastern half of the continent.
In general, the scientific question addressed and the method to answer it are entirely relevant and sound. Among other interesting results, I find particularly original and smart the evaluation of the capacity of the model to predict the life cycle of heat waves, taking into account the relative time of forecast issuance and heatwave initiation. However, in my view three aspects need to be revised or further elaborated before the manuscript can be accepted. These points, detailed below, concern first the structure of the manuscript which requires improvements, then the case of the summer of 2010 which needs to be further discussed, and also some missing specifications in the description of the method.

Response:  We are very grateful for your comments, views and improvement suggestions, they really give a lot for the manuscript. It was especially nice to receive such encouraging comments about Figure 6! For point-by-point responses to parts that require improvement, please see below.

**Specific comments**
1- Paper organization:
I find the organization of the manuscript rather clumsy, in particular the discussion part that dwells into strategies of adaptation to heat, thereby repeating or elaborating some elements of the introduction. Additionally, it sometimes go quite far (too far!) into details when it comes to adaptation and preparedness measures, keeping in mind that the core of the paper is an evaluation of heat wave forecast skill.
The conclusion reads like a shorter repetition of the discussion.
Finally, the part of the discussion providing avenues for enhancing heat wave forecast skill (hence more aligned with the main work of this paper) refers very vaguely to "additional bias techniques" and cites  a list of (sub-)seasonal forecast predictors without indicating in which manner they could contribute to refine the heat wave forecasts.

Response: Thank you for this comment. We are going to edit the Discussion strongly. We shall leave out the detailed adaptation and preparation measures, and the additional bias techniques. We shall focus more on discussing the skill of these forecasts shown in the manuscript. We shall also give more attention to the results of Figure 6, the heat wave life cycle figure, and the significance of our results considering the weight of the summer 2010 heat wave.
We would also like to mention here that we are going to change the title to be: "The probabilistic **skill** of Extended-Range Heat wave forecasts over Europe" due to the comment by another referee:
*"The word "usefulness" in the title made me as a reader expect something (more related to climate services) that is not shown in the study.-- "*

2- Impact of the summer of 2010 in eastern Europe / western Russia:
That summer was characterized by a particularly long lasting heat wave over that region, and this is well reflected in your manuscript. Yet, that summer of 2010 seems to 'contaminate' your results and conclusions  : it is particularly obvious when comparing your fig. 4g with 4i (and 4j with 4l), keeping fig.

2a in mind. Without that particular summer of 2010, most of the skill over Europe is gone in weeks 3 and 4. Could you elaborate on this, and discuss the significance of your results considering the huge weight of that event ?

Response: A good point, thank you. We now plotted figures 1, 2, 4, and 6 (see below) to compare our results for the period 2000-2019 with and without 2010. Figure 1 gives a spatial distribution, with 1 °C intervals, for the threshold of the heat wave days for the period 2000-2019 with (Figure 1, first column) and without 2010 (Figure 1,middle column). One can see that if summer 2010 is excluded, the north-west-southeast gradient is very close to that for the whole 2000-2019 period. The last column shows the impacts of including 2010: in most of the western and the southern Europe the difference is ±0.1°C, while in the eastern and north-eastern parts of Europe the impact is mostly between 0 and +0.55 °C, except for the very northern Fennoscandia where the impact is between -0.2 and 0 °C.

[Figure]

**Figure 1: The lower thresholds of heat wave days: the 90th percentile of the 5-day moving average temperature in summers 2000-2019 (first column) and in summers 2000-2009 and 2011-2019 (i.e., 2000-2019 excluding 2010, middle column) of the ERA5 reanalyses (a and b), and (d,e,g,h,j,k,m, and n) of the ensembles of the ECMWF's hindcasts in different forecast weeks. The last column shows the difference between these two.**

Figure 2b indicates that if the summer 2010 is excluded, other years (e.g., 2014) appear in eastern Europe / western Russia, compared to Fig. 2a, and the duration of the longest period of heat wave day get shorter there. In Fig 2d especially in those areas where 2010 had the longest period of heat wave days, excluding it means an increase in the number of periods with heat wave days, as the 10% of the hottest days are now distributed to a larger number of events.

[Figure]

**Figure 2: The duration and the year of the longest period of heat wave days defined from the ERA5 reanalysis data of (a) summers 2000-2019 and (b) summers 2000-2009 and 2011-2019 (i.e., 2000-2019 excluding 2010) (marked as 0-19), and the number of periods with heat wave days (b) in the ERA5 reanalyses during (c) 2000-2019 and (d) 2000-2009 and 2011-2019 (i.e., 2000-2019 excluding 2010).**

In the revised Figure 4 (below), the last column now shows the BSS of the hindcasts excluding the summer 2010. Leaving out 2010 actually seems to have less impact than leaving out, in each grid point, the summer with the longest heat wave (the middle column). For example, in Finland the skill remains for the third week. Also the southeast parts of the study domain seem to remain with skill. **Hence: the best of the skill seems to come from the longest period of heat wave days, whether it was the 2010 heat wave or a heat wave of some other year.**

**Brier skill scores (*BSS*) of the probabilistic heat wave days forecasts, *p***

[Figure]

Figure 4: Brier Skill Scores (BSS) of the probabilistic heat wave days forecasts, p, during all summers 2000-2019 (first column), in hindcasts excluding the summer with the longest period of heat wave days (middle column), and in hindcasts excluding the summer 2010 (last column). The statistical occurrence p=0.1 for heat wave days were used as the reference forecasts. The dotted areas show where BSS is greater than zero with the false discovery rate no more than 10%.

As you might have noticed in Figure 4, we have decided to leave out the "summer with the longest heat wave" as another referee pointed out that:

*Figure 4, ll. 288 – 295: While I think excluding the summers with the longest heat waves gives a good idea of how strongly the overall skill of the forecasts is influenced by these events, I don't think we can learn much from the skill for just the summer with the longest heat wave. While it seems to be in line with the conclusions from the right column in Fig. 4, I would argue that all the middle column might be telling us is that the reference forecast is particularly bad when you choose to basically look at one event alone (meaning ot in the BS is 1 most of the time and thus the BS of climatology, i.e. pt = 0.1, gets very high, because now your climatological forecast is not reliable anymore). Unless of course you recalculate the 90th percentile using only one summer, which is obviously problematic (representativeness), too.*

[Figure]

**Figure 7 (DATA EXCLUDING YEAR 2010): The forecasted probabilities of heat wave days shown for days that (in ERA5) were 21 to 1 days before the heat wave, the 1ˢᵗ to 35ᵗʰ heat wave day during the heat wave, and 1 to 21 days after the heat wave with lead times of a) one week, b) two weeks, c) three weeks, and d) four weeks. Dashed green boxes indicate forecasts where, at the time of issuance, a heat wave in that grid point was about to begin within a week. Solid green boxes indicate forecasts where, at the time of issuance, a heat wave was already ongoing in that grid point. The boxplots (as in Fig.5) include all forecast data across except summer 2010 the European region at each land-grid point.**

We also plotted the heat wave life cycle figure (Figure 6) without year 2010, here as Figure 7.
Leaving out year 2010 removes most of the very longest heat waves, i.e, with lengths above 28 days.
However, the same way as with including the year 2010, in the forecast weeks 1-3: there is still signal of enhanced accuracy in forecasting prolonged (here several weeks long) heat waves at the time that the heat wave had initiated prior to the forecast issuance.

These figures showing the effect of excluding year 2010 will be included in the revised manuscript.

3- Method:
 L.115-117: It is not very clear if your take the lead time into account when computing the 90th $TEC5d$ . For example, when computing this value for, say, July 1st : do you compute one single value by pooling together all the hindcast members that include the sequence July 1st- July 5th (regardless of the start date) ?
Response: No, we did not.

Or instead, do you compute different percentile values according to the lead time (i.e. one value if July 1st is part of week 1, another one if it is part of week 2 etc.) ?

Response: Yes, we did.

From my understanding, the former strategy allows a larger statistical sample to compute percentiles, but on the other hand, there is a potential impact of lead-time dependency. The latter one seems more accurate from this point of view but then of course the sample size is smaller. I guess the "lead time dependent climatology" indicated on l.125 refers to this strategy.
Response: Yes, the latter one is the one we used.

Additionally, you should specify the range of start dates used in the method. I believe this would help understand how you computed percentiles for the very first days of June in particular. In other words, did you include hindcasts initialized in early May, to ensure a homogeneous sample size throughout all summer days ?
Response: No, we did not.

Or did you only consider the first days of June as part of "week 1" lead time)?
Response: Yes we did.

Could you clarify (and potentially discuss) these method points in the manuscript ? Maybe include a schematic or a table if needed.

Response: Thank you for the comments. Sure, we will add a Table (see below) with explanations.

[Figure]

Table 1A. Table showing details of the investigated hindcasts. Each row contains one run, altogether 12 runs. The first red boxes on each row show the initiation date of the hindcasts, which are same for all years 2000-2019. The data of days marked with red are used for lead time 1 week, blue for 2 weeks, yellow for 3 weeks, and grey for 4 weeks. The forecast data used for the forecast weeks were partially overlapping due to the use of 5-days moving averages with forward-looking window: the forecast week 1 used data of days 1 to 11, the forecast week 2 data of days 8 to 18, forecast week 3 data of days 15 to 25, and forecast week 4 data of days 22 to 32. The data used for two lead times are here marked with two colours. Note: for lead time 1 week we used data of 12 runs, for lead time 2 weeks we used data of 11 runs, for lead time 3 weeks we used data of 10 runs, and for lead time 4 weeks we used data of 9 runs (of years 2000-2019).

**Technical corrections:**

L.179 "by change" => "by chance" (?)

Response: Thank you, we shall correct this as suggested.

L.196 Do you mean "how early a heat wave becomes..." or "how early heat wave days become" ?

Response: Thanks for noticing this, we mean "how early heat wave days become", we shall correct this.

L.248-250 : OK, but this sounds more like a rephrasing of what precedes than a new information.

Response: Thank you, we shall remove this sentence on lines 248-250.

L. 271-273: Nice result for week 3 but it would be fair to remind that the sample size of week 3 forecasts with p>0.5 is probably very small, considering Figure 3a.  So I think this result should be considered with a pinch of salt.

Response: Yes, thank you, we agree. We shall point this out more clearly.

Figure 1: I would recommend to display the 4 ECMWF maps as "bias wrt. ERA5", ie plotting the difference "ECMWF minus ERA5". The associated comment L. 213 would be more convincing.

Response: Thank you for this comment. We shall consider this, as technically this is easy to make.

L. 268 and elsewhere : Choose between "Figure 3(b)" or Figure "3b" (even better: remove one of them) and choose also between "Fig." and "Figure" .

Response: Thank you. We shall edit this.

Non-existing "Figure 3d" shows on L. 272. Better remove it, since it seems quite obvious that you keep commenting Figure 3b here.

Response: Thank you for mentioning this. We shall correct 3d to 3b.

L. 294: Typo (?) . I was expecting : "Eastern Europe (summer 2010)"

Response: Thank you for mentioning this. We shall correct 2018 to 2010.

L.319-326: Ok but this part is very unpleasant to read. Please rephrase by not repeating the exact same sentence 4 times !

Response: A good point, thank you. We shall edit this to be:
In occasions the forecasted probability for heat wave days was low ($p$<0.33), heat wave days occurred in 2% (lead time one week), 7% (lead time two weeks), 10% (lead time four weeks), or 11% (lead time four weeks) of cases. Moreover, in occasions the forecasted probability for heat wave days was intermediate (0.33≤$p$≤0.66), heat wave days occurred in 45% (lead time one week), 39% (lead time two weeks), 30% (lead time four weeks), or 28% (lead time four weeks) of cases. In occasions the forecasted probability for heat wave days was high ($p$>0.66), heat wave days occurred in 86% (lead time one week), 68% (lead time two weeks), 67% (lead time four weeks), or 38% (lead time four weeks) of cases.

L. 406: heat-health action plans (2 typos)

Response: Thank you for mentioning this. We shall correct this as you suggest.

---

## Author Comment (AC2)

**MS. Ref. No.: NHESS-2024-75 "The usefulness of Extended-Range Probabilistic Forecasts for Heat wave forecasts in Europe"**
**Natural Hazards and Earth System Sciences**

We appreciate your constructive comments as they help to enhance the quality of our manuscript. The following are point-by-point answers in blue colour:

**Referee #2 (RC2):**

This study investigates the probabilistic skill of extended-range forecasts of mildly extreme land temperatures over Europe. It shows that these forecasts are overall reliable out to the third forecast week, but, except for Eastern/Southeastern Europe, do not significantly differ in skill from a much simpler climatological forecast. The skill of the forecasts appears to be strongly enhanced by the most long-lasting events. Excluding these events results in reduced skill over almost all of Europe. An analysis of the evolution of skill throughout the life cycle of the heat wave indicates that the models capture the persistence of anomalous temperatures well, whereas the onset and end of the events seem more difficult to predict. The study presents a relevant contribution to the field of evaluation of extended- range/subseasonal prediction for potentially impactful events. While previous studies have considered the prediction skill of the same extended-range forecasts for extreme temperatures before, this study adds a thorough assessment of the probabilistic skill of the forecasts by using some well-documented methods and scores (which facilitates comparability) and providing some more non-standard ways of looking at the prediction skill. Assessing the probabilistic instead of the deterministic skill of the forecasts is arguably much more important in the extended range, since their uncertainty is large, but the information in the spread of the ensemble could still make the forecasts reliable. The employed methods are sensible and the skill analysis for the heat wave life cycle is innovative and a highlight of the paper. I do, however, not entirely agree with the way the study is framed. The title implies that the study assesses the usefulness of the forecasts, which I don't think it does. The authors also stress the health impacts of heat waves a lot, which is of course a good motivation to investigate the skill at predicting heat extremes, but the study does not include any analysis that links the forecasts to health impacts/heat stress in any way. Furthermore, some of the methods should be explained and motivated more clearly. Finally, the writing could be made more concise in many places. I provide more detailed comments below.

Response: We are very grateful for your comments, views and improvement suggestions, which all give a lot for the manuscript. It was especially nice that you mentioned the skill analysis for the heat wave life cycle as the highlight of the paper! For point-by-point responses to parts that require improvement, please see below.

**Major remarks:**
1. The word "usefulness" in the title made me as a reader expect something (more related to climate services) that is not shown in the study. The usefulness of a forecast can only be determined by involving its user(s), which also means that the forecast, in most cases, will be useful only to some but not others. Furthermore, for a forecast to be useful, skill (which is what I think the article is actually focused on) is just one of many requirements. So, unless the analysis is extended significantly and involves this component, I suggest changing the word "usefulness" to something else here (maybe "skill" would be most accurate).

Response: Thank you for this comment, we agree, and we will change the title to be:
"The probabilistic skill of Extended-Range Heat wave forecasts over Europe"

2. There is a lot of text concerning health impacts/risks of heat in the discussion (ll. 403 – 443). While I don't generally disagree with anything that is written about this, I don't think it deserves the amount of space it is given in the discussion, given there is no direct relationship with the presented results. The

study investigates the probabilistic skill of summer forecasts for mildly extreme (dry bulb) temperatures and the discussion should focus on this aspect. The authors offer an explanation for why they use the temperature measures that they use, and I think it is fair to focus on these, but there is evidence indicating that other measures of temperature are more strongly related to heat stress (involving radiation, humidity, wind) and thus more suitable for measuring health risk/impact of heat events, see e.g. Di Napoli et al. (2019), McGregor & Vanos (2018). Thus, I would suggest removing the too detailed discussion of health impacts of heat from Section 4.

Response: Thank you, we will remove the too detailed discussion of health impacts of heat from Section 4, as suggested.

Alternatively, if the focus on health impacts should be kept, I suggest considering the use of other, possibly more heat-stress-related, metrics.

Di Napoli, C., F. Pappenberger, and H. L. Cloke, 2019: Verification of Heat Stress Thresholds for a Health-Based Heat-Wave Definition. J. Appl. Meteor. Climatol., 58, 1177–1194, https://doi.org/10.1175/JAMC-D-18-0246.1.

McGregor, Glenn R., and Jennifer K. Vanos, 2018: Heat: a primer for public health researchers, Public Health, 161, 138-146, https://doi.org/10.1016/j.puhe.2017.11.0053.

If the current focus of the paper is kept, I think the discussion needs to be revised strongly. As mentioned above, the part ll. 403 – 443 seems very detached from the results of the study right now. The remaining text in Section 4 (ll. 374 – 401) is more of a summary and is to a large degree repeated in Section 5 (where it belongs, in my opinion). I think this part could be used better to discuss the implications, the potential and the limitations of your study (as you do in ll. 444 -452), see below for some suggestions:

• One question that I wonder about when seeing the results (although it is beyond the scope of the paper to answer this finally): Could it be that the forecasts are generally too persistent and thus lucky when a long-lasting heat wave happens, or do they actually "know" when to persist temperatures? In other words, are they right for the right reasons? The fact that the exclusion of the most long-lasting events basically removes all remaining skill from the week 3 & 4 forecasts makes me think that they might just have been lucky. Also, your Figure 6 could be interpreted further with this question in mind.

Response: Yes, thanks. Due to this we also have in Figure 6 shown how and how early the model forecasts the ending of heatwaves. It is a good idea to add interpretation about that here.

• You mention climate change in the discussion (l. 433). Against the backdrop of climate change, what do your results mean? Are we expecting better forecasts because we will see more (and potentially longer-lasting) heat extremes? Or might the predictability of these events also change?

Response: We mention climate change and the intensification of heat waves here as that means that heat wave forecasts are expected to be needed in the future as well. We will clarify this.

• Parts of your manuscript suggest that you would like to link this to the applicability of extended-range forecasts in early warnings of heat waves (e.g. l. 391). Could you elaborate on what your results mean, e.g. for an agency that would want to implement these forecasts for early warnings? Is the skill sufficient? Can the presented aggregation over large geographical areas (5˚ x 2˚) be useful in some way? Where can the forecasts contribute and where can't they, keeping in mind that they are ok at predicting the persistence but not so good at predicting the onset far in advance?

Response: Thanks, this is a good question. We agree that at least this way used the forecasts do not predict the onset of a heat wave 3-4 weeks beforehand, but as it shows to capture the persistence of

heat waves well. As the longest heat waves have high impact, even the smallest piece of information about them is reasonable to take into account.

4. Since you try to address usefulness/applicability of the forecasts, it could be a good idea to assess reliability on a regional ("grid-point") level in addition to the BSS (Fig. 4). The reason is that reliability can be linked better to decision-making, see Weisheimer & Palmer (2014). Their paper shows a simple method of categorizing forecasts by the slope (and its uncertainty) of their reliability curve into 5 categories. This would address the usefulness aspect at least to some degree and could be a nice addition to the current results.

Weisheimer, A., and T. N. Palmer, 2014: On the reliability of seasonal climate forecasts, J. R. Soc. Interface, 11: 20131162. http://dx.doi.org/10.1098/rsif.2013.1162

Response: Thanks for the good idea. However, as we now changed the title of the manuscript (see Major remark 1.) to "The probabilistic skill of Extended-Range Heat wave forecasts over Europe" we would like to stay with the current analysis, and maybe rather mention this in the discussion.

5. In general, Section 3 could use some additional explanations to make the results of the analysis easier to grasp for the reader. Generally, at the beginning of each subsection (3.X.), provide one sentence on why we're seeing this plot now and what it's supposed to tell us (like you do in ll. 276 – 277).

Response: Yes, thank you. This certainly is a good suggestion, and we shall add explanations in the beginning of each subsection (3.X).

More specifically:
i. Section 3.5: Since this is not a very standard form of presenting forecast skill (at least not one I'm familiar with), I suggest explaining the reason for showing the skill in this form. I get the feeling it is relatively closely related to the reliability diagram. In what way does it differ/provide extra information? What can we learn from this way of looking at the forecasts? As a reference for the reader, give an example of what a good and a poor forecast would look like if displayed in this way (as you do in the part with the reliability diagram). A bit more information on this could also aid the interpretation of the next plot.

Response: In the reliability diagram (Fig 3b) the ERA5-based temperature data is used only as either no hot day (0) or hot day (1). However, this figure 5 shows also how near or far away the ERA5-based temperature was from the threshold of a hot day (the $90^{th}$ percentile, the grey line). It is a good idea to add more information to explain and we will do that.

ii. Figure 6, Section 3.6: I consider the life cycle plot a highlight of the manuscript, but it contains a lot of information, so I think it deserves a more thorough discussion (and to be picked up in Section 4!). One thing I find particularly noteworthy in this figure is that, while there seems to be an upward trend in the forecast probabilities leading up and into the heat waves, the highest probability class (p > 0.66) is only really predicted when the heat wave is already present in the initialization of the forecast.

Response: Thank you for this comment. We agree, and we shall add a more thorough discussion of this figure!

**Minor comments:**

Title
"forecast" is used twice, could maybe reformulate?

Response: Yes, (see Major remark 1.) we will change the title to be:

"The probabilistic skill of Extended-Range Heat wave forecasts over Europe"

Intro

l. 26: 'intense and prolonged heat waves during the third forecast weeks' The study doesn't really address intensity, so the first part of this should be removed. I also think it would be more accurate to say that persistence of heat/extreme temperatures seem to have a higher level of predictability. The current sentence suggests that the forecasts are generally (onset, duration, intensity, ending) better for strong events.

Response: Thank you for this comment. We shall remove word "intense" here. And we shall edit the sentence to be: "Nonetheless, persistence of heat waves seems to have a higher level of predictability--"

l. 28: one sentence linking back the results of the study to the motivation (early warning systems) would round off the introduction a bit more.

Response: Thank you this is a good idea. We shall add a sentence here.

l. 32: 'in future' to 'in the future'

Response: Thank you, we shall correct this by adding word "the" here.

l. 37: 'particularly so in urban areas' can be removed since there is no relation of this to the question the study addresses.

Response: Thank you, we shall remove 'particularly so in urban areas'.

ll. 46 – 54: I think it should be mentioned here that high (dry bulb) temperature is only one factor in heat stress, see references I provide above.

Response: Thanks, we shall mention it here.

ll. 55 – 63: I understand this paragraph as a motivation to consider the prediction of longer-term averages of temperature. If that's the case, be more explicit about it and say that due to the above reasons there could be value in considering the prediction of these averages. This could also be related to the fact that longer aggregations might be better predictable, see e.g.

Toth, Z. and R. Buizza (2019). "Weather Forecasting: What Sets the Forecast Skill Horizon?" In: Sub-Seasonal to Seasonal Prediction: The Gap Between Weather and Climate Forecasting. Ed. by A. Robertson and F. Vitart. 1st. Elsevier. Chap. Chapter 2, 17 –45.

Response: Yes, thank you, we agree and will add this motivation and reference.

ll. 59 – 62: I don't see the relevance of this with regards to the study. Can be removed.

Response: Ok, we shall remove suggested lines 59-62.

ll. 64 – 75: This fits more into the general motivation of the study at the beginning of the intro (potentially in a shortened form)

Response: Thank you for the comment. We shall shorten this for some parts, but for the introduction we would keep the shortened ll. 64 – 75 where it was, as this paragraph ends with telling about the

current length of heat wave forecasts in Europe, and from that it is good for us to continue in the next paragraph about the skill of extended range long heat wave forecasts.

l. 64: 'alleviate the tendency towards more frequent and intense heat waves' I don't understand what this means.

Response: Thank you for the comment. Here the idea of this sentence was, that we mention the importance of mitigating the ongoing climate change to mitigate the intensifying of heat waves as they are projected to intensify the more the higher the atmospheric greenhouse gas concentration. However, it is not necessary to have this sentence here, so we can remove it (and make this chapter a bit shorter).

ll. 82 – 85: work out more clearly what your study is adding and providing beyond what has been done previously. Stress the probabilistic nature of the forecasts that you are evaluating and the analysis of the 'heat wave skill life cycle'

Response: Thanks, we shall add this information here.

ll. 86 – 89: This is already mentioned in ll. 55 – 62 and does not need to be repeated here

Response: Thank you, we shall remove lines 55-62.

l. 91: change 'forecasts' to 'hindcasts' or 're-forecasts'

Response: Thank you, we hall use 'hindcasts'.

l. 94/95: These two sentences seem a bit redundant as they are now. Can you be a bit more specific in guiding the reader through the paper here?

Response: Thanks, we shall specify here that we investigate the forecasts' reliability, BSS and the model's ability to forecast the life cycle of the heatwaves, taking into account the relative time of forecast issuance and heatwave initiation.

**Methods**

l. 96: The word 'Materials' seems a bit off in the context of the study. Maybe 'Data' is more appropriate?

Response: Yes, good point, thank you, we shall change the word 'Materials' to 'Data'.

ll. 100 – 101: This could maybe be formulated more carefully. The skill of the hindcasts gives an indication of the skill of the forecasting system, but it is not necessarily the same (as you point out in ll. 126 – 134, so maybe merge these sentences).

Response: Yes, good point, thank you. We shall merge these parts as you suggest.

ll. 101: Meaning all forecasts initialized during JJA (which includes forecast and verification for September days) or all with verification dates in JJA?

Response: Meaning all forecasts initialized AND having verification dates in JJA. Hence, we did not include hindcasts initialized in early May (or those reaching September). We shall clarify this by a data Table 1A (below) and explanations.

[Figure]

Table 1A. Table showing details of the investigated hindcasts. Each row contains one run, altogether 12 runs. The first red boxes on each row show the initiation date of the hindcasts, which are same for all years 2000-2019. The data of days marked with red are used for lead time 1 week, blue for 2 weeks, yellow for 3 weeks, and grey for 4 weeks. The forecast data used for the forecast weeks were partially overlapping due to the use of 5-days moving averages with forward-looking window: the forecast week 1 used data of days 1 to 11, the forecast week 2 data of days 8 to 18, forecast week 3 data of days 15 to 25, and forecast week 4 data of days 22 to 32. The data used for two lead times are here marked with two colours. Note: for lead time 1 week we used data of 12 runs, for lead time 2 weeks we used data of 11 runs, for lead time 3 weeks we used data of 10 runs, and for lead time 4 weeks we used data of 9 runs (of years 2000-2019).

ll. 102 – 104 & l. 106: What is the reason for only using Monday initializations instead of all available ones?

Response: Thank you for this comment. The reason for using only the Monday initialization, was to have each date in each lead time only once to avoid autocorrelation. We could have chosen to use the Thursday runs only as well.
We shall add this information to the manuscript as well.

l. 109: The ECMWF (re-)forecasts are run at higher horizontal resolution up to day 15 and then re-initialized at lower resolution from day 15 to 46.

Response: Thank you, we shall add this information here.

ll. 112 - : I suggest starting with defining heat wave days for the verification since the verification data is simpler (it only has one time dimension). Then you only have to explain how you handle the extra time dimension (lead time) in the hindcasts.

Response: Thank you, good point we shall rearrange this as you suggest.

l. 117: Is this the 90th percentile of all (summer) days under consideration or for each calendar day individually?

Response: The 90th percentile is of all (summer) days under consideration.

l. 125: bias → frequency bias

Response: Thanks, sure we can correct this.

ll.127 – 134: Maybe this could be re-structured a bit because it seems to be going back and forth between saying the hindcast ensemble is large enough to get an idea of the forecasting system's skill and saying it is not.

Response: Yes, thank you. We shall re-structure this and merge it with ll 100-101, as suggested in a comment above.

l. 134: Another important difference between the skill shown in the study and the skill of the actual forecasting system is that in forecast mode, there is no information about the future, while you are using all years (including the evaluated one) when defining the percentiles. This is likely to lead to an overestimation of the skill. To simulate this setting, a leave-one-year-out cross validation could be employed. I'm not requesting the authors to do this, but I think it should be pointed out in addition.

Response: Thank you for the comment. We shall mention this.

ll. 141 – 142: This sentence sounds like it is stating the obvious. Maybe better to say something like: "A single below-threshold day between two heat wave days was nevertheless classified as a heat wave day."

Response: Thank you, we shall edit this.

ll. 144 – 149: see comment on Table 1 below.

Response: Yes, thank you, we shall do as suggested, i.e., remove the old Table 1 and write the information about it here in section 2.2 as text only.

l. 168: do you mean "define this period as the summer containing the longest heat wave"? Is the entire summer taken out or just the period of the longest heat wave?

Response: We meant the entire summer, however, this will be removed (see comment for Figure 4 ll. 288-295)

ll. 175 - 1178: Could you provide a more detailed description of how the bootstrap resampling procedure works?

Response: Yes, sure we can add here a more detailed description of how the bootstrap resampling procedure works.

l. 179: "change" → "chance"

Response: Thanks, the word 'change' is corrected to 'chance'.

ll. 182 – 183: Explain in a few words how this procedure works.

Response: Yes, sure we can here explain in a few words how this procedure works.

ll. 184 – 190: This seems to be better placed in the part where you explain how you generate a probabilistic forecast from the ensemble.

Response: Thank you, good idea, we shall move it there as you suggest.

ll. 191 – 192: Why these categories? They seem rather arbitrary. Are they used somewhere, which would justify considering them here?

Response: Thank you for the comment, we shall clarify this by: "We conducted verification of heat wave day forecasts across all grid points in Europe based on forecasted probabilities falling within the ranges of here defined as low: p<0.33, intermediate: 0.33≤p≤0.66, and high: p>0.66."

l. 196: "a heat wave days become discernible" I don't understand this, please reformulate

Response: Yes, thank you. We shall edit this to: "(To investigate how early) heat wave days appear (in the forecasts)"

ll. 203 – 205: This part is a bit difficult to understand (especially before having seen Figure 6). Maybe reformulate this.

Response: Thanks, yes me can reformulate this to make it more clear.

Results
ll. 210 – 211: I think the information in this sentence is redundant here and already given where it is relevant.

Response: Thank you. We shall remove this sentence on lines 210-211.

ll. 219 – 226, Table 1: What do you conclude from these numbers and how is this relevant for the forecasts or even their skill? Maybe this could rather become part of the method section (2.2.) if the point is to justify the definition of heat waves using the 5-day mean. To me, it wasn't clear why I'm seeing the table at this point in the paper. Since the information in the table is also entirely contained in the text, you could consider removing the table.

Response: Thank you, this is a good idea to remove the (old) Table 1 from here and include the text from here to the method section(2.2). We shall do that.

ll. 231 – 237: The same as the above comment applies to this subsection. This is just looking at ERA5, so it has nothing to do with the forecasts. I suggest moving this to Section 2 where the heat wave definition or the exclusion of the longest events is described. Alternatively, dedicate a short section at the beginning of Section 3 to the analysis that only deals with ERA5.

Response: Thank you for this comment. We haven't decided yet which alternative to adopt but shall either move this to Section 2 or have it in a short section at the beginning of Section 3.

l. 245: I think its noteworthy that this is not valid the other way around. You aren't claiming that, but I think it helps a reader who might be less familiar with the details of forecast verification to stress that sharpness is a property of the forecasts alone, i.e. 90% forecasts with p = 0 and 10% with p = 1 does not directly imply a perfect forecast (i.e. sharpness is a necessary but not a sufficient condition).

Response: A good point, thank you. We shall mention this here.

l. 259: match → equal

Response: Thank you, we shall edit 'match' to 'equal'.

l. 267: by → with

Response: Thank you, we shall edit 'by' to 'with'.

l. 268: can drop the parentheses, it is mentioned in the sentence before.

Response: Thank you, we shall drop the parentheses.

ll. 270 – 271: "reliability remained higher than that achieved by climatology alone" →
this statement cannot be true since by the way you define climatology (i.e. without

leaving the validation year out) it has perfect reliability by definition (but no resolution).

Response: Thank you for this comment. We shall remove this sentence.

ll. 271 – 273: I think there is a mix-up here between the "no skill-line" and the reliability of climatology. Climatology (as defined here) has perfect reliability, so no forecast can possibly have better reliability. It does, however, not have any resolution (it predicts p = 0.1 in all instances) and so its BS is higher than 0. If points lie above the "no skill-line" it means that they contribute positively to the BSS with climatology as reference. This is comparing the BS of the forecast to the BS of climatology, not just the reliability. For details see:

Mason, S. J., 2004: On Using "Climatology" as a Reference Strategy in the Brier and Ranked Probability Skill Scores. Mon. Wea. Rev., 132, 1891–1895, https://doi.org/10.1175/1520-0493(2004)132<1891:OUCAAR>2.0.CO;2.

Response: Thanks. We shall check this and correct the mix-up.

l. 280/281: "the predictions […] demonstrates" → "the forecasts […] demonstrate"

Response: Thank you, we shall make the suggested editing.

l. 282: superior to the reference forecast → different from 0

Response: Thanks, we shall edit this to "greater than 0".

l. 284: as before, here you basically say "BSS remains better than the reference forecast" while what you mean is that the BSS remains above zero, or alternatively, the forecasts remain better than the reference.

Response: Thank you, good point. We shall edit this to "forecast remaining better than the reference forecast".

Figure 4, ll. 288 – 295: While I think excluding the summers with the longest heat waves gives a good idea of how strongly the overall skill of the forecasts is influenced by these events, I don't think we can learn much from the skill for just the summer with the longest heat wave. While it seems to be in line with the conclusions from the right column in Fig. 4, I would argue that all the middle column might be telling us is that the reference forecast is particularly bad when you choose to basically look at one event alone (meaning ot in the BS is 1 most of the time and thus the BS of climatology, i.e. pt = 0.1, gets very high, because now your climatological forecast is not reliable anymore). Unless of course you recalculate the 90th percentile using only one summer, which is obviously problematic (representativeness), too.

Response: Ok, good point. We will remove the column showing the skill for just the summer with the longest heat wave.

l. 317: refer back to Figure 3a?

Response: Ok, this is visible both in Fig 5 and Fig. 3a, so we shall add here "which was also visible in Fig. 3a".

Figure 5: Why is the total n (sum of n for all 3 categories) for each subplot different? Shouldn't this add up to the total number of forecast days within each forecast week times the number of considered grid points?

Response: As we did not include hindcasts initialized in early May (or those reaching September), there was actually the largest amount of data for forecast week 1 and smallest amount of data for forecast week 4. We shall clarify this by a data Table 1A (below) and explanations.

[Figure]

Table 1A. Table showing details of the investigated hindcasts. Each row contains one run, altogether 12 runs. The first red boxes on each row show the initiation date of the hindcasts, which are same for all years 2000-2019. The data of days marked with red are used for lead time 1 week, blue for 2 weeks, yellow for 3 weeks, and grey for 4 weeks. Note: for lead time 1 week we used data of 12 runs, for lead time 2 weeks we used data of 11 runs, for lead time 3 weeks we used data of 10 runs, and for lead time 4 weeks we used data of 9 runs (of years 2000-2019).

Ll. 334 – 335: I don't quite understand what is meant by the notches here. The second sentence rather belongs into the results with a description of where we see this in the plot and what it implies.

Response: Thanks, we shall clarify and edit this.

Section 3.6: I find it a bit confusing that the results are described from the longest to the shortest lead time here, when throughout the rest of the paper, the description starts with week 1. Maybe an option to invert the order?

Response: Thanks. Good point, we shall invert the order.

l. 349: no need to put the "green box" in quotation marks.

Response: Thanks, we shall remove the quotation marks from the "green box".

ll. 368 – 372 (caption Figure 6): what are the limits of the box plots? Same as in Figure 5, i.e. interquartile range and whiskers for 5th and 95th percentile?

Response: Yes, as in Fig 5, i.e., the horizontal line dividing each box into two parts shows the median of the data; the ends of the box show the lower and upper quartiles; and the whiskers indicate the 5th and 95th percentiles of the data in each group.

l. 448: "as introduced to result from"; I don't understand what this means.
l. 451: "the land-atmosphere interaction" → "land-atmosphere interactions"
ll. 444 – 452: Could you be more specific about how this could be used to refine the forecasts?

Response: Thank you for these remarks. This part of the discussion might be excluded as it is not about our methods and hence it will not be further edited.

ll. 458 – 462: This is almost an exact repetition of ll. 383 – 387. Keep it only in one place (I'd suggest Section 5).

Response: Thank you, we keep this only in one place.

ll. 473 – 478: Like the aforementioned part of the discussion (ll. 403 – 443), this paragraph seems very detached from the core results of the paper. Rather end the conclusions with some outlook for future work and how it could be continued to make it even more relevant in the context you bring up here.

Response: Thank you for the comment, we shall exclude this part (ll. 203-443) and end the conclusion with some outlook for future work.

---

## Referee Report (RR1)

**Review nhess-2024-75-2 "The probabilistic skill of Extended-Range Heat wave forecasts in Europe"**

I appreciate the effort that you have put into addressing my comments and I think parts of the manuscript have improved. As stated in the previous round, I find the results interesting and relevant and the study worth publishing. In general, however, large parts of the paper are still difficult to comprehend and lack explanations and motivations. For clarity and readability, the manuscript requires some major revisions. I try to outline the major issues that I still see and what I think you could do about them below.

**Major general remarks:**

**Introduction**

In my opinion, the paragraphs in the intro need to be linked more, both to each other and more explicitly to the specifics of the paper. It is currently difficult to see why certain things are mentioned where they are mentioned.

For instance, ll. 41 – 45 go into some details on the effects of heat waves on buildings in Northern Europe. When I read this for the first time, it seemed like an overly specific example, but I think that what you might mean to do here is to argue for why it makes sense to look at longer term averages of temperatures (l. 44: "heating of buildings has been observed to take 5 – 6 days"). If this is the case, please be more explicit about it. This applies to other parts of the intro, too.

Another example is the transition from l. 59 to l. 60. You nicely explain why earlywarnings systems are needed and what general systems are in place. If you add one sentence on why "even earlier" warning systems (i.e. based on subseasonal forecasts) could be beneficial, you make a transition to introducing the extended range forecasts.

**Methods**

I like the new Table 1, which makes it very easy to grasp the structure of the data being used for the verification. I also appreciate that you took my suggestion of moving the part that relates solely to the verification data into the methods section. However, I think this section's comprehensibility would benefit strongly from some major re-structuring. I think most parts of the text are there, and they could be re-arranged and slightly re-written. I recommend the following to improve the readability, but I acknowledge that this is somewhat subjective:

Try to go from the most basic to the details. To me, the definition of a heat wave (day) is the most essential and basic piece of information in the context of the paper\*. It should be put first in the method section along with your explanation for why this is a meaningful threshold (ll. 161 – 166). Mention that with your definition, you transform a continuous variable (temperature) into a binary variable, which is your forecast target. Then you could introduce what you assume as reality/ground truth/verification (namely, ERA5, maybe commenting quickly on shortcomings of reanalysis in representing reality) and go into some detail on what heat waves defined in this manner look(ed) like and how robust the definition is (ll. 167 – 175). Here, you could also use what is now Fig 2 (and potentially Fig. 1a or even 1a-c) and discuss the "outlier" 2010 and why it deserves some special attention.

I would only then move to the forecasts. Introduce the model and the ensemble system set-up. Then explain how the "extra" time dimension (lead time) is treated when defining heat waves days and that thresholds are defined with respect to the model climatology. Explain how you go from an ensemble forecast (essentially a "collection" of deterministic forecasts) to a probability forecast. Finally, you can talk about how to verify them (current section 2.3).

\*I'd picture something like ll.73 – 74 but introducing variables such as  $T^{5d}$ ,  $T^{5d,90}$  and saying that you only include land areas.

Figure 1: From the current text, I'm not sure why this figure is shown (except maybe a-c which could be used as I indicated above). The main point of using a model-dependent threshold to define heat waves in the forecasts is to avoid any issues with differences in this threshold between models and re-analysis. So, what do you conclude from the fact that they are basically the same? What would be different if you saw that the  $T^{5d,90}$  were very different in forecasts vs. re-analysis? If I'm just missing the point here, please give more concrete conclusions from this figure in the manuscript.

**Discussion**

I think the discussion should be extended. It is fair to say that the results are in line with other studies, but is this expected or unexpected given the employed methodologies/approaches? What are possible limitations of your study, where could it be extended (you mention this a bit in ll. 407 – 409) and why is it nevertheless important as it is? I like that you dedicate a section to the potential added value of probabilistic forecasts, but in its current form this section is for the most part a short literature review (ll. 411 – 424), which is better placed in the Intro. Ll. 423 – 429 go in the right direction in my opinion. Additionally, are there maybe examples of events where it is thought that even earlier warnings would have been beneficial in mitigating some of the effects of a heat wave? How important are things like the spatial resolution and temporal aggregation in this context? You mention the relevance of 5-6 day temperature averages for Northern Europe, but is this valid in other countries that might have a completely different building stock?

**Further comments:**

Title: mix of capitalized and lower-case words

- l. 19: in extended range ightarrow in the extended range
- l. 27: "persistence [...] seem to have"  $\rightarrow$  "persistence [...] seems to have a"

ll. 69 – 70: I think this last sentence might be better placed in the discussion.

l. 75: "have"  $\rightarrow$  "has"

l. 106: "initiation"  $\rightarrow$  "initialization"

l. 122 "capture"  $\rightarrow$  "skillfully predict"

l. 141: "forecasting in the model's climatology" I don't quite understand how this is meant. Is it just to say that a heat wave in the forecast is defined relative to the forecast model's climatology? Maybe you could reformulate.

l.144 (also see my earlier comment from the first round on l. 134): As you correctly say here, you are implicitly bias-correcting the hindcasts. Since you are not leaving out the year for which you forecast (this year would not be available in a real forecast, because it has not happened yet), this is a better correction than you could ever have access to in reality. This will lead to an overestimation of the skill (although the larger ensemble in forecast mode might counteract this). You do show that the 90th percentile does not change much in absolute terms when leaving out the most severe events, so it's reasonable to assume that the effect on skill is not huge, but this does not mean that there is no effect. In conclusion, I think you should mention this point, as it is generally agreed upon that S2S hindcast verification should be done in a leave-one-out manner.

l. 175: It would be ideal to end this paragraph with a sentence about what you conclude from these statistics.

ll. 211 –220: would be good to add a subscript or something to distinguish the forecast probability from the base rate (currently, you call both p).

l. 220: "base rate of p"  $\rightarrow$  "base rate  $p_b$ " (or whatever else you will call the base rate)

l. 236: "the BSS *n* times (here n = 5000)"  $\rightarrow$  "the BSS 5000 times" (no need to define *n* if you never use it again)

l. 243: "the FDR controls for the expected proportion of false discoveries" I don't quite understand what this means. Isn't the FDR the proportion of false discoveries? Could you maybe reformulate?

l. 245: Thanks for adding an explanation on the B-H procedure. It is still not entirely clear to me why this is necessary in addition to the p-value adjustment. Could you elaborate?

l. 267: "a heat wave days"  $\rightarrow$  "heat wave days"

l. 273: "shorter forecast weeks"  $\rightarrow$  "shorter lead times"

l. 275: It could be noted here again that there are a lot less samples in the higher probability bins (expect for maybe lead time 1 week, bin 0.9 – 1), so those points are a lot more uncertain.

l. 280: "of the all hindcasts"  $\rightarrow$  "of all the hindcasts"

l. 295: add comma after "In the second week"

l. 299: what do you take from this analysis? Are the results strongly influenced by the longest heat wave (or 2010)? It looks to me like the skill in weeks 2 – 4 is systematically lower when the longest heat waves are excluded with only few exceptions. In most areas differences are small, so maybe it only really matters in Eastern Europe/Russia (skill goes from being significant to being not significant). Also, I think the left and middle rows might be enough to show. Or what extra information do you gain from excluding 2010 everywhere? If you decide to show it, you should discuss it more.

l. 300: "In the Figure 4"  $\rightarrow$  "In Figure 4"

l. 305: two commas at the end of the line

Section 3.3: I appreciate that you included some more background on why to look at the forecasts in this way. However, I am still a bit confused about what we learn from this plot, so I think it would help to add what you are concluding from this analysis at the end of the section. You say the point is "to assess the severity of the over- or underforecasts". So, based on these plots, how severe is it for different lead times? Is it possible to relate this type of evaluation to any of the fundamental properties of a forecast, e.g. is it related to discrimination or resolution (in the forecast verification sense)?

I'm also thinking about the bins/categories you use. Now, they are basically: extremely elevated likelihood (p > 0.66), strongly elevated likelihood (0.33 ) and everything from moderately elevated likelihood to lower likelihood (<math>p < 0.33) for having a heat wave. I think it would make this plot a lot more interesting if the forecasts were split at p = 0.1 and p was expressed relative to the base rate. Below that threshold, forecasts indicate a lower-than-normal likelihood of a heat wave and above, they indicate a higher likelihood. You could have one "basically normal/climatological likelihood" category with something like 0.05 and one below (reduced likelihood) and one above (increased likelihood). Also, the statement "we are 5 times more likely than normal to have a heat wave in week X" has a very different psychological effect than saying "the chances of having a heat wave in week X are 50%", which makes me think it might be more interesting to see <math>p relative to the base rate.

L. 321 – 324: I think this is of little help in understanding the plot, because your perfect forecast would only ever issue p=0 or p=1, which is a very hypothetical situation. Could you rather say what a good (but not infinitely sharp) vs. a poor or no-skill forecast would look like?

l. 322: "p<0.33 be"  $\rightarrow$  "p < 0.33 would be"

l. 339: "amount"  $\rightarrow$  "fraction"

l. 343: "the relative time of forecast issuance and heat wave initiation" do you mean the forecast initialization (date) relative to the onset of the heat wave?

l. 344: "corresponsive"  $\rightarrow$  "corresponding"?

l. 357 – 362: Some statements from ll. 353 – 356 are repeated verbatim here. Is it an option to wrap these into one, as in: "In both forecast week 1 and 2, there is …"

l. 371: "indicating ongoing heat"  $\rightarrow$  "indicating an ongoing heat wave"

caption Figure 6: add what *n* and the width of the boxes mean. Also see my comment on Section 3.3/Figure 5: maybe an option to express p relative to the base rate?

Figure 7 and ll. 384 – 387: I'm not sure this figure is needed. What extra information does it provide? The main difference is that there remains hardly any data in the "29+ day inside the heat wave" categories, which just shows that there is basically no event inside the sample that is as long-lasting as 2010. But this does not really tell us any more about the forecasts. Since the differences are small in the categories that are well populated, you could just mention that the differences are negligible.

ll. 393 – 397: This is a short recap of the method. Why is it relevant in the context of the discussion here? Is there some relation (similarities/differences) to the methods used in the papers you cite in l. 398?

l. 403: "the best of the forecast skill seems to come from the longest period of heat wave days" Do you mean that the skill in forecasting heat waves decreases when excluding the event?

l. 439: would add here that this significant skill largely vanishes when 2010 is excluded (Fig. 4)

l. 448: "its"  $\rightarrow$  "heat wave occurrence"

l. 448: "further"  $\rightarrow$  remove

---

## Referee Report (RR2)

Review for "The probabilistic skill of Extended-Range Heat wave forecasts over Europe", by Korhonen et al.

**General comments:**

I acknowledge the efforts of the authors to tackle all the issues raised by the reviewers, and improve the manuscript accordingly.
I still find the lengthy discussion part in lines 411-424 quite odd. In my view, it would better fit the introduction section, since it does not discuss the results found but rather explains the motivation to carry out this study. However, I would not make a strong case for another round of revision, and I suggest the paper be accepted for publication.
I am listing below a few typos found in the revised manuscript.

**Typos:**
L. 280: typo "all the hindcasts"
Fig.4 : typo in the title of the second column: "longest"
L.349: typo "if the forecasts were perfectly.."

---

## Referee Report (RR3)

**Review nhess-2024-75-3 "The probabilistic skill of Extended-Range Heat wave forecasts in Europe"**

Thank you for the responses and clarifications to my comments and questions. As before, I think the analysis you carried out is interesting and relevant. I also think that after your latest revisions, the method and all technical aspects of your analysis have become very clear. I still think the discussion part and to some degree the introduction could be improved in terms of readability, but I see these as minor points and leave it up to the authors to decide how much they would like to address these.

**Some more detailed comments:**

ll. 43 – 52: This paragraph is overly specific. It is fair to cite your studies here since they are somewhat relevant to an aspect of this paper, but the level of detail is unnecessary in my opinion, as it distracts the reader from the main message. E.g.,

> l. 43: Add something like "As an example, during heat waves, apartments… ". You go from listing heat wave impacts very generally to something extremely specific (buildings in the Nordics) here.

> l. 45 – 46: I think this level of detail is not necessary here and this sentence could be dropped.

> ll. 49 – 50: This sentence could be dropped, too. I don't see it adding any relevant reason for why to consider longer term temperature averages. If anything, doesn't it make the case weaker?

ll. 84 – 86: To me, this sentence just says, that a lot of people (in which case there should be more than one example study) have written about this topic. I think it can be dropped.

ll. 90 – 102: I think this paragraph gets slightly too specific on some details of the analysis and could rather give a more general overview over what you're doing in the paper (for which the motivation should be covered at this point of the intro).

ll .95 – 97: You have motivated this already earlier (ll. 43 – 52), I don't think it needs to be motivated again.

Figure 2, ll. 176 – 180: In (d) you have one year less, so doesn't it make more sense to show events per year instead of total number of events?

l .186: initiated → initialized

l. 192: "which are *the* same"

l.195 – 196: "runs *per summer/year*"

l. 203 – 205: Could replace these two sentences by: "We therefore *arbitrarily* decided to use only the Monday runs."

l. 217: "*daily* mean"

ll. 226 – 227: I think this could be dropped, since it has been clearly defined already.

l. 230: Could mention here that this difference does not influence your verification because you consider model-specific thresholds.

l. 242: Remove "Moreover,"

l. 255: remove ", *p,*"

ll. 258 – 259: "probability (of the heat wave day)" → "probability of a heat wave day"

l. 259: good one :)

l. 281: "greater *than* zero"

l. 283: "the results" → would rather say the significance/importance of the results

ll. 316 – 317: "for lead times of 2 weeks (and longer) there are far fewer samples" → I would say this is true for all lead times. Except maybe week 1 $p = 0.9$, although also those are far fewer than $p = 0.1$.

ll. 356 – 357: could maybe explicitly say that here, you transform the probabilistic forecast to a categorical one. I find this quite appealing by the way, because I think this is a much more likely scenario of how such a forecast would be used in practice. I think it could also be fair to mention that.

Figure 5: If you want to, there is a lot more that you could unpack from this figure and the way you look at the forecasts here (i.e., as categorical forecasts). While in this figure, it is easy to see what the outcome was, given a certain forecast (category) was issued, it could be quite insightful to see what the forecast was, given a certain outcome (heat wave/no heat wave). You can actually do this (approximately) with the numbers in the plot. For instance, the forecasts for all lead times have a high likelihood (decreasing with lead time but > 90% for all) $p(f_0|o_0)$ of forecasting no heat wave when there turned out to be no heat wave, i.e. they have high specificity. However, when you wrap up the $(0.33 < p < 0.66)$ and $(p > .66)$ categories into one "forecast of heat wave" category, the true positive rate (sensitivity) $p(f_1|o_1)$ drops from 86% in week 1 to 19% in week 4. In addition, the likelihood $p(f_0|o_1)$ of forecasting "no heat wave" when there was actually a heat wave, goes up from 18% in week 1 to almost 99% in week 4. This means that if a forecast at longer lead times indicates "heat wave", there are good chances there will be one, but if it shows "no heat wave", you shouldn't rely on there being none. In my

eyes, this could be quite important information to someone designing an early-warning system. I leave it up to you, but in my opinion something like this would make a very nice addition to the paper and it could add some points that could be addressed in the discussion.

l. 415: "(days 29..35)" → "(days 29 to 35)"?

l. 494: Does ERF stand for "extended-range forecast"? Don't think this is defined anywhere.

---

## Author Response (AR2)

**Authors' response to both Referee #1 and Referee #2 (Report 2)**

**Referee #1 (RC1):**

**Review for "The probabilistic skill of Extended-Range Heat wave forecasts over Europe", by Korhonen et al.**

**General comments:**

I acknowledge the efforts of the authors to tackle all the issues raised by the reviewers, and improve the manuscript accordingly.
I still find the lengthy discussion part in lines 411-424 quite odd. In my view, it would better fit the introduction section, since it does not discuss the results found but rather explains the motivation to carry out this study. However, I would not make a strong case for another round of revision, and I suggest the paper be accepted for publication.
I am listing below a few typos found in the revised manuscript.

**Typos:**
L. 280: typo "all the hindcasts"
Fig.4 : typo in the title of the second column: "longest"
L.349: typo "if the forecasts were perfectly.."

Response: Thank you for these comments. We have corrected the mentioned typos and moved lines 411-424 in a shortened form to the Introduction section.

**Referee #2 (RC2):**

**Review nhess-2024-75-2 "The probabilistic skill of Extended-Range Heat wave forecasts in Europe"**

I appreciate the effort that you have put into addressing my comments and I think parts of the manuscript have improved. As stated in the previous round, I find the results interesting and relevant and the study worth publishing. In general, however, large parts of the paper are still difficult to comprehend and lack explanations and motivations. For clarity and readability, the manuscript requires some major revisions. I try to outline the major issues that I still see and what I think you could do about them below.

We appreciate your constructive comments as they further helped to enhance the quality of our manuscript. The following are point-by-point answers in blue colour:

Major general remarks:

*Introduction*

In my opinion, the paragraphs in the intro need to be linked more, both to each other and more explicitly to the specifics of the paper. It is currently difficult to see why certain things are mentioned where they are mentioned.

For instance, ll. 41 – 45 go into some details on the effects of heat waves on buildings in Northern Europe. When I read this for the first time, it seemed like an overly specific example, but I think that what you might mean to do here is to argue for why it makes sense to look at longer term averages of temperatures (l. 44: "heating of buildings has been observed to take 5 – 6 days"). If this is the case, please be more explicit about it. This applies to other parts of the intro, too.

Another example is the transition from l. 59 to l. 60. You nicely explain why early-warnings systems are needed and what general systems are in place. If you add one sentence on why "even earlier" warning systems (i.e. based on subseasonal forecasts) could be beneficial, you make a transition to introducing the extended range forecasts.

Thank you for these comments. We have now made several modifications to the introduction to link the paragraphs to each other, point out why the things are mentioned where they are, and motivate the use of 5 days mean temperature.

First, we added to the end of the first paragraph of the Introduction (now lines 35-37): "This growing occurrence of heat waves underscores the urgent need to understand their dynamics and improve forecasting methods, especially for prolonged events with severe impacts."

l.41-45 was edited as (now lines 46-52): "In Northern Europe, where apartments are typically not equipped with mechanical cooling systems, the thermal inertia of buildings plays a critical role. For instance, a Finnish study observed that buildings required 5-6 days to reach overheating conditions, highlighting the importance of the 5-day mean temperature as a predictor for indoor heat stress (Velashjerdi Farahani 2024a). In not well-insulated buildings and/or light structures, such as wooden ones, the warm-up time can be significantly shorter, often only 1–2 days. These findings emphasize the relevance of forecasting tools capable of predicting not only the occurrence but also the persistence of heat waves."

l. 51 was connected to the heat-health action plans by (now lines 57-59): "Recognizing this, many countries in Europe and other parts of the world have developed heat-health action plans over the past 20 years to mitigate heat-related health risks (Kotharkar et al. 2022; Martinez et al. 2022)."

l. 59-60 we added (now lines 68-70): "Extending these lead times could significantly enhance preparedness by allowing for earlier adaptive measures and better resource allocation, particularly for prolonged heat waves."

and we start the next paragraph by (now lines 72-73): "Sub-seasonal forecasts, which cover the extended range of 2 weeks to 1 month, offer a promising avenue for improving early warning systems."

Parts of the discussion ll. 411-424 have been placed into the third paragraph of the introduction, now lines 59-65 (and these lines have been completely removed from the discussion):
"As health effects of heat exposure occur quickly, at the same day or a few days lag (Baccini et al. 2008), it is imperative that the protection measures are implemented rapidly when a potentially dangerous heat wave is forecasted. However, organization of the response measures requires coordination of actions between many stakeholders and distribution of workforce, equipment, and other resources, which take time."

*Methods*

I like the new Table 1, which makes it very easy to grasp the structure of the data being used for the verification. I also appreciate that you took my suggestion of moving the part that relates solely to the verification data into the methods section. However, I think this section's comprehensibility would benefit strongly from some major re-structuring. I think most parts of the text are there, and they could be re-arranged and slightly re-written. I recommend the following to improve the readability, but I acknowledge that this is somewhat subjective:

Try to go from the most basic to the details. To me, the definition of a heat wave (day) is the most essential and basic piece of information in the context of the paper*. It should be put first in the method section along with your explanation for why this is a meaningful threshold (ll. 161 – 166). Mention that with your definition, you transform a continuous variable (temperature) into a binary variable, which is your forecast target.

Then you could introduce what you assume as reality/ground truth/verification (namely, ERA5, maybe commenting quickly on shortcomings of reanalysis in representing reality) and go into some detail on what heat waves defined in this manner look(ed) like and how robust the definition is (ll. 167 – 175). Here, you could also use what is now Fig 2 (and potentially Fig. 1a or even 1a-c) and discuss the "outlier" 2010 and why it deserves some special attention.

I would only then move to the forecasts. Introduce the model and the ensemble system set-up. Then explain how the "extra" time dimension (lead time) is treated when defining heat waves days and that thresholds are defined with respect to the model climatology. Explain how you go from an ensemble forecast (essentially a "collection" of deterministic forecasts) to a probability forecast. Finally, you can talk about how to verify them (current section 2.3).

*I'd picture something like ll.73 – 74 but introducing variables such as $T5d$, $T5d,90$ and saying that you only include land areas.

Thank you for these recommendations to improve readability of the Methods. We have now strongly re-arranged and where needed re-written the text so that:
The Definition of the heat wave days is now first, in Section 2.1.
Thereafter, in Section 2.2. we introduce the ERA5 data (together with its Fig 1a and Fig 2).
Next, in Section 2.3 we move on to the hindcasts and probabilistic forecasts, and Section 2.4 introduced the used skill scores.

Figure 1: From the current text, I'm not sure why this figure is shown (except maybe a-c which could be used as I indicated above). The main point of using a model-dependent threshold to define heat waves in the forecasts is to avoid any issues with differences in this threshold between models and re-analysis. So, what do you conclude from the fact that they are basically the same? What would be different if you saw that the $T5d,90$ were very different in forecasts vs. re-analysis? If I'm just missing the point here, please give more concrete conclusions from this figure in the manuscript.

Thank you for this comment. We have now included contents of the Figure 1 in the following text added in Section 2.3.2 (this text was related to the comment on l.144 also) now lines 245-250:
"In the verification, the forecast model-based probability of a heat wave day, p, was compared to the observed heat wave days (Section 2.2.1) derived from the ERA5 dataset. Since we used the data from the entire period (years 2000–2019) to define the heat wave day thresholds, we may achieve an overestimation of the forecast skill in the verification compared to using a leave-one-out method (in which one year is excluded at a time from the dataset when defining the threshold). However, as shown in the last column of Figure 1, excluding even the most extreme year has only a minimal impact on the threshold definition. Therefore, it is reasonable to assume that the effect on the skill is not substantial."

*Discussion*

I think the discussion should be extended. It is fair to say that the results are in line with other studies, but is this expected or unexpected given the employed methodologies/approaches? What are possible limitations of your study, where could it be extended (you mention this a bit in ll. 407 – 409) and why is it nevertheless important as it is? I like that you dedicate a section to the potential added value of probabilistic forecasts, but in its current form this section is for the most part a short literature review (ll. 411 – 424), which is better placed in the Intro. Ll. 423 – 429 go in the right direction in my opinion. Additionally, are there maybe examples of events where it is thought that even earlier warnings would have been beneficial in mitigating some of the effects of a heat wave? How important are things like the spatial resolution and temporal aggregation in this context? You mention the relevance of 5-6 day temperature averages for Northern Europe, but is this valid in other countries that might have a completely different building stock?

Thank you for these comments. We edited the Discussion for many parts.

The first paragraph of the Discussion (now lines 438-443) we edited to be:
"We examined the skill of hindcasts of the ECMWF in forecasting the probability of heat wave days over Europe 1 to 4 weeks ahead. The assessed hindcasts demonstrated varying levels of accuracy across different regions, and decreasing levels with increasing forecasting lead times, which is in line with many earlier studies, e.g., Wulff and Domeisen (2019), and Pyrina and Domeisen (2023). This outcome could be seen as expected, as we employed the same forecasting model and verification region as in these previous works. However, our method for determining the probability of a heat wave day was novel, providing a fresh perspective that sets our study apart from earlier research using the same model and verification region."

Additionally, parts of the discussion ll. 411-424 have been placed into the introduction (and these lines have been completely removed from the discussion).

Further, considering question: *You mention the relevance of 5-6 day temperature averages for Northern Europe, but is this valid in other countries that might have a completely different building stock?*, we have now added the following information to the Introduction, now lines 45-50:
"The warm-up time related to outdoor temperature depends on building properties (U-value, ventilation airflow rate, and thermal mass of buildings). In typical Nordic well-insulated apartment buildings, e.g., brick and concrete apartment buildings, the warm-up time is 5- 6 days. In not well-insulated buildings and/or light structures, e.g., wooden structures, the warm-up time is much shorter. In those cases, it could be only 1- 2 days."

Considering question: *are there maybe examples of events where it is thought that even earlier warnings would have been beneficial in mitigating some of the effects of a heat wave?*
We have now added to the discussion, now lines 463-473:
"To the knowledge of the authors, there has been no published research on how warning lead time contributes to the effectiveness of heat-health warning systems. However, considering the short lag between heat exposure and worsening of health conditions, extending warning lead times from the current level of few days is acknowledged to be valuable to public health, as prevention and emergency measures need to be in place and operational at the onset of a hazardous heat event (WHO 2021). Organization of the measures, such as communication campaigns, establishing cooling centers, arrangements to protect vulnerable population groups, and ensuring adequate supply and distribution of workforce, equipment, and other resources, require time and would benefit from receiving early warnings 1–2 weeks ahead, particularly because heat waves often occur at times when organizations and services are already short-staffed due to summer holiday season. Longer lead time is especially important regarding exceptionally severe and prolonged hot periods, which challenge the functioning of society on a wider scale and may require large-scale interagency and even

transboundary response. The likelihood for these types of events can be expected to increase in Europe as climate change progresses."

Regarding to possible limitations of our study:
We acknowledge in the methods section (now lines 209-213) that the ensemble size of the hindcast dataset (11 members) differs from that of the current operational forecasts (101 members). Using operational forecasts would involve significantly more effort and extend beyond the scope of this study. Moreover, operational forecasts might cover a shorter historical period, limiting their utility for our analysis.
Another limitation is that the ensemble spread could have been examined in greater detail, as briefly mentioned in the discussion (now lines 451-453 ). However, since the probabilistic forecast utilizes the entire ensemble, the spread is inherently accounted for in the analysis.
Further possible limitation of our study is that we used only one model (ECMWF), as we write in the conclusion (now lines 490-492): "-- future research could investigate at which stage of the heat wave development extended-range weather forecast models in general, not only the specific model system considered here, begin to predict heat wave occurrence, --"
The selected definition "heat wave day" also might be a limitation, however as we mention in section 2.1. (now lines 114-115) "Our definition of heat wave days is meaningful as it aligns with thresholds commonly used in epidemiological studies on heat-related health effects, --"
Another limitation is that this study focuses on forecasting heat wave days; however, it is important to note that what ultimately concerns people are the impacts, such as health effects. Predicting these impacts is beyond the scope of this work.

Further comments:

Title: mix of capitalized and lower-case words
Thank you for pointing this out. We now corrected this to title case, i.e., The Probabilistic Skill of Extended-Range Heat Wave Forecasts Over Europe.

l. 19: in extended range → in the extended range
l. 27: "persistence [...] seem to have" → "persistence [...] seems to have a"
Thank you for these corrections, we have edited the text as suggested.

ll. 69 – 70: I think this last sentence might be better placed in the discussion.
Thank you for the comment. We reformulated this and left it in the introduction, now lines 86-88: "In theory and practice, probabilistic forecasts have been shown to contain more information and should be more valuable to users than categorical, deterministic forecasts (Murphy 1977, Richardson 2001), though their practical utility depends on users' ability to incorporate such information into decisions (e.g., Lopez & Haines 2017; Ramos et al. 2013)."

l. 75: "have" → "has"
l. 106: "initiation" → "initialization"
l. 122 "capture" → "skillfully predict"
Thank you for these three corrections, we have edited the text as suggested.

l. 141: "forecasting in the model's climatology" I don't quite understand how this is meant. Is it just to say that a heat wave in the forecast is defined relative to the forecast model's climatology? Maybe you could reformulate.
Thank you for this remark. We have reformulated this as: Hence, a heat wave in the forecast is defined relative to the forecast model's climatology.

l.144 (also see my earlier comment from the first round on l. 134): As you correctly say here, you are implicitly bias-correcting the hindcasts. Since you are not leaving out the year for which you forecast (this year would not be available in a real forecast, because it has not happened yet), this is a better correction than you could ever have access to in reality. This will lead to an overestimation of the skill (although the larger ensemble in forecast mode might counteract this). You do show that the 90th percentile does not change much in absolute terms when leaving out the most severe events, so it's reasonable to assume that the effect on skill is not huge, but this does not mean that there is no effect. In conclusion, I think you should mention this point, as it is generally agreed upon that S2S hindcast verification should be done in a leave-one-out manner.

Thank you for this remark. We have added this text here in Section 2.3.2, now lines 245-250:
"In the verification, the forecast model-based probability of a heat wave day, p, was compared to the observed heat wave days (Section 2.2.1) derived from the ERA5 dataset. Since we used the data from the entire period (years 2000–2019) to define the heat wave day thresholds, we may achieve an overestimation of the forecast skill in the verification compared to using a leave-one-out method (in which one year is excluded at a time from the dataset when defining the threshold). However, as shown in the last column of Figure 1, excluding even the most extreme year has only a minimal impact on the threshold definition. Therefore, it is reasonable to assume that the effect on the skill is not substantial."

l. 175: It would be ideal to end this paragraph with a sentence about what you conclude from these statistics.

Thank you for this remark. We added here (now lines 146-148): "These statistics show that the 5-day moving average definition covers nearly all longer heat wave events (such as 3- to 4-day heat waves), but only a portion of shorter ones (1- to 2-day heat waves). This indicates that the 5-day moving average is particularly useful for identifying sustained heat wave events.".

ll. 211 –220: would be good to add a subscript or something to distinguish the forecast probability from the base rate (currently, you call both $p$).
l. 220: "base rate of $p$" → "base rate $p_b$" (or whatever else you will call the base rate)
l. 236: "the BSS $n$ times (here $n = 5000$)" → "the BSS 5000 times" (no need to define $n$ if you never use it again)

Thank you for these improvement suggestions, we have edited the text as suggested. Hopefully it is now clearer.

l. 243: "the FDR controls for the expected proportion of false discoveries" I don't quite understand what this means. Isn't the FDR the proportion of false discoveries? Could you maybe reformulate?
l. 245: Thanks for adding an explanation on the B-H procedure. It is still not entirely clear to me why this is necessary in addition to the p-value adjustment. Could you elaborate?

Thank you for these remarks. We reformulated these and now it is hopefully more clear that the FDR is more like a concept and the B-H is one of the possible procedures to implement it.

l. 267: "a heat wave days" → "heat wave days"
l. 273: "shorter forecast weeks" → "shorter lead times"

Thank you for these corrections, we have edited the text as suggested.

l. 275: It could be noted here again that there are a lot less samples in the higher probability bins (expect for maybe lead time 1 week, bin 0.9 – 1), so those points are a lot more uncertain.

Thank you for pointing this out. We continued the sentence by (now lines 316-317)"; however, it should be noted that for lead times of 2 weeks (and longer) there are far fewer samples in the higher probability bins, making these points considerably more uncertain."

l. 280: "of the all hindcasts" → "of all the hindcasts"

l. 295: add comma after "In the second week"
Thank you for these corrections, we have edited the text as suggested.

l. 299: what do you take from this analysis? Are the results strongly influenced by the longest heat wave (or 2010)? It looks to me like the skill in weeks 2 – 4 is systematically lower when the longest heat waves are excluded with only few exceptions. In most areas differences are small, so maybe it only really matters in Eastern Europe/Russia (skill goes from being significant to being not significant). Also, I think the left and middle rows might be enough to show. Or what extra information do you gain from excluding 2010 everywhere? If you decide to show it, you should discuss it more.
Thank you for this comment. We added (now lines 344-345) "These results suggest that the skill in forecasting heat waves decreased when excluding the longest period of heat wave days, whether it was the 2010 heat wave or a heat wave from another year."

l. 300: "In the Figure 4" → "In Figure 4"
l. 305: two commas at the end of the line
Thank you for these corrections, we have edited the text as suggested.

Section 3.3: I appreciate that you included some more background on why to look at the forecasts in this way. However, I am still a bit confused about what we learn from this plot, so I think it would help to add what you are concluding from this analysis at the end of the section. You say the point is "to assess the severity of the over- or under-forecasts". So, based on these plots, how severe is it for different lead times? Is it possible to relate this type of evaluation to any of the fundamental properties of a forecast, e.g. is it related to discrimination or resolution (in the forecast verification sense)?
I'm also thinking about the bins/categories you use. Now, they are basically: extremely elevated likelihood ($p > 0.66$), strongly elevated likelihood ($0.33 < p < 0.66$) and everything from moderately elevated likelihood to lower likelihood ($p < 0.33$) for having a heat wave. I think it would make this plot a lot more interesting if the forecasts were split at $p = 0.1$ and $p$ was expressed relative to the base rate. Below that threshold, forecasts indicate a lower-than-normal likelihood of a heat wave and above, they indicate a higher likelihood. You could have one "basically normal/climatological likelihood" category with something like $0.05 < p < 0.2$ and one below (reduced likelihood) and one above (increased likelihood). Also, the statement "we are 5 times more likely than normal to have a heat wave in week X" has a very different psychological effect than saying "the chances of having a heat wave in week X are 50%", which makes me think it might be more interesting to see $p$ relative to the base rate.
Thank you for this comment. To clarify how the severity of the over – or underforecasts can be assessed, we have now added about Fig. 5 (now lines 382-385):
"It should be noted that Figure 5 also shows how often forecasts were followed by a heat wave or *near-heat wave* conditions (e.g., temperatures exceeding the 85th percentile) in the ERA5 dataset. For instance, in situations where p > 0.66, temperatures surpassing the 85th percentile (rather than the 90th percentile) occurred even in 95% (lead time one week), 78% (lead time two weeks), 74% (lead time four weeks), or 44% (lead time four weeks) of cases."
For the threshold p=0.1 we added to the text (now lines 378-380):
"Additionally, $p < 0.33$ provides a good indication that a heat wave is unlikely. Based on the data, the lower the $p$ (below 0.33), the less likely a heat wave is to occur, as, e.g., in occasions the $p < 0.1$ (no figure), heat wave days occurred only in 1% (lead time one week), 4% (lead time two weeks), 6% (lead time three weeks), or 8% (lead time four weeks) of cases."

L. 321 – 324: I think this is of little help in understanding the plot, because your perfect forecast would only ever issue $p$=0 or $p$=1, which is a very hypothetical situation. Could you rather say what a good (but not infinitely sharp) vs. a poor or no-skill forecast would look like?
Thank you, good point. We have now added here (as lines 368-370):
"All in all, the forecast skill improves when more of the data points in p < 0.33 fall below the grey line,

and those in p > 0.66 are above the grey line. At a glance, forecast week 1 (Fig. 5a) appears to have good skill, while forecast week 4 (Fig. 5d) shows relatively poor skill."

l. 322: "p<0.33 be" → "$p < 0.33$ would be"
l. 339: "amount" → "fraction"
Thank you for these corrections, we have edited the text as suggested.

l. 343: "the relative time of forecast issuance and heat wave initiation" do you mean the forecast initialization (date) relative to the onset of the heat wave?
Thank you for pointing this out. We mean indeed the forecast initialization (date) relative to the onset of the heat wave, and we have now edited the text accordingly.

l. 344: "corresponsive" → "corresponding"?
Thank you for this remark, we have edited the text as suggested.

l. 357 – 362: Some statements from ll. 353 – 356 are repeated verbatim here. Is it an option to wrap these into one, as in: "In both forecast week 1 and 2, there is ..."
Thank you for this comment. We have edited this as (now lines 406-410):
"In heat wave day forecasts both one week in advance (Figure 6a) and two weeks in advance (Figure 6b), the forecasts show clearly higher p for days within the heat wave than outside, especially for the forecasts which are in the green boxes indicating that the heat wave was just starting or already underway when these forecasts were issued. Additionally, there is some overestimation, particularly 1-2 days before or after the heat waves indicating slight inaccuracy in forecasting the exact day of the start and ending of the heat wave."

l. 371: "indicating ongoing heat" → "indicating an ongoing heat wave"
Thank you for this remark, we have edited the text as suggested.

caption Figure 6: add what *n* and the width of the boxes mean. Also see my comment on Section 3.3/Figure 5: maybe an option to express p relative to the base rate?
Thank you for these comments. We have added the explanation for *n* and the width of the boxes. Regarding the suggestion to express p relative to the base rate, we believe that the current presentation of Figures 5 and 6 works best for our purposes, and therefore, we have decided not to make changes in this regard.

Figure 7 and ll. 384 – 387: I'm not sure this figure is needed. What extra information does it provide? The main difference is that there remains hardly any data in the "29+ day inside the heat wave" categories, which just shows that there is basically no event inside the sample that is as long-lasting as 2010. But this does not really tell us any more about the forecasts. Since the differences are small in the categories that are well populated, you could just mention that the differences are negligible.
Thank you for this comment. We agree and we have now put this Figure 7 as Supplementary Material 1 and we have also included here in the text: "Thus, the differences remain negligible."

ll. 393 – 397: This is a short recap of the method. Why is it relevant in the context of the discussion here? Is there some relation (similarities/differences) to the methods used in the papers you cite in l. 398?
Thank you for this remark. We have removed from here the short recap of the method and further edited this paragraph to be (now lines 438-443):
"We examined the skill of hindcasts of the ECMWF in forecasting the probability of heat wave days over Europe 1 to 4 weeks ahead. The assessed hindcasts demonstrated varying levels of accuracy across different regions, and decreasing levels with increasing forecasting lead times, which is in line with many earlier studies, e.g., Wulff and Domeisen (2019), and Pyrina and Domeisen (2023). This

outcome could be seen as expected, as we employed the same forecasting model and verification region as in these previous works. However, our method for determining the probability of a heat wave day was novel, providing a fresh perspective that sets our study apart from earlier research using the same model and verification region."

l. 403: "the best of the forecast skill seems to come from the longest period of heat wave days" Do you mean that the skill in forecasting heat waves decreases when excluding the event?
Thank you for this comment. Yes, we edited this to be (now lines 447-448): "We found that the skill in forecasting heat waves decreased when excluding the longest period of heat wave days, whether it was the 2010 heat wave or a heat wave of some other year."

l. 439: would add here that this significant skill largely vanishes when 2010 is excluded (Fig. 4)
Thank you for the comment. We here remove the word "Eastern" and leave the text as (now lines482-483) "--in the forecast weeks 3-4: statistically significantly better skill than the reference forecast only in some grid points across South-Eastern Europe" it mentions that significant skill is only in *some* grid points across South-Eastern Europe, and those are the areas in which statistically significantly better skill than the reference forecast remained even when 2010 was excluded.

l. 448: "its" → "heat wave occurrence"
l. 448: "further" → remove
Thank you for these remarks, we have edited the text as suggested.

---

## Author Response (AR3)

**Authors' response to Referee #2 (Report 3)**

**Referee #2 (RC2):**

Review nhess-2024-75-3 "The probabilistic skill of Extended-Range Heat wave forecasts in Europe" Thank you for the responses and clarifications to my comments and questions. As before, I think the analysis you carried out is interesting and relevant. I also think that after your latest revisions, the method and all technical aspects of your analysis have become very clear. I still think the discussion part and to some degree the introduction could be improved in terms of readability, but I see these as minor points and leave it up to the authors to decide how much they would like to address these.

We appreciate your constructive comments as they further helped to enhance the quality of our manuscript. The following are point-by-point answers in blue colour:

Some more detailed comments:

ll. 43 – 52: This paragraph is overly specific. It is fair to cite your studies here since they are somewhat relevant to an aspect of this paper, but the level of detail is unnecessary in my opinion, as it distracts the reader from the main message. E.g.,
l. 43: Add something like "As an example, during heat waves, apartments… ". You go from listing heat wave impacts very generally to something extremely specific (buildings in the Nordics) here.
l. 45 – 46: I think this level of detail is not necessary here and this sentence could be dropped.
ll. 49 – 50: This sentence could be dropped, too. I don't see it adding any relevant reason for why to consider longer term temperature averages. If anything, doesn't it make the case weaker?
Thank you for these comments, we have removed these suggested parts:
l. 45-46 "The warm-up time of buildings related to outdoor temperature depends on building properties (U-value, ventilation airflow rate, and thermal mass of buildings)."
and ll.49-50 "In not well-insulated buildings and/or light structures, such as wooden ones, the warm-up time can be significantly shorter, often only 1–2 days."

ll. 84 – 86: To me, this sentence just says, that a lot of people (in which case there should be more than one example study) have written about this topic. I think it can be dropped.
Thank you for this comment, we have dropped this sentence: "There is a large literature in statistics and decision analysis on the use of probabilistic information in so-called decision making under uncertainty (e.g. Clemen 1996)."

ll. 90 – 102: I think this paragraph gets slightly too specific on some details of the analysis and could rather give a more general overview over what you're doing in the paper (for which the motivation should be covered at this point of the intro).
ll .95 – 97: You have motivated this already earlier (ll. 43 – 52), I don't think it needs to be motivated again.
Thank you for these comments, we have dropped this suggested sentence from ll 95-97: "Moreover, in an empirical study conducted in Finland, indoor temperatures were found to be more strongly correlated with outdoor 5-day moving average temperature than with average temperatures of a few days only, suggesting impacts of building's thermal inertia (Velashjerdi Farahani et al., 2024b)."

Figure 2, ll. 176 – 180: In (d) you have one year less, so doesn't it make more sense to show events per year instead of total number of events?
Thank you for this remark, however, here we are leaving this comparison as it is.

l .186: initiated → initialized

l. 192: "which are *the* same"

l.195 – 196: "runs *per summer/year*"

l. 203 – 205: Could replace these two sentences by: "We therefore *arbitrarily* decided to use only the Monday runs."

l. 217: "*daily* mean"

ll. 226 – 227: I think this could be dropped, since it has been clearly defined already.

l. 230: Could mention here that this difference does not influence your verification because you consider model-specific thresholds.

l. 242: Remove "Moreover,"

l. 255: remove ", *p,*"

ll. 258 – 259: "probability (of the heat wave day)" → "probability of a heat wave day"

l. 259: good one :)

l. 281: "greater *than* zero"

*Thank you for these remarks, we have corrected these as suggested.*

l. 283: "the results" → would rather say the significance/importance of the results

*Thank you for this remark, we agree, and we have edited this to be "the significance of the results".*

ll. 316 – 317: "for lead times of 2 weeks (and longer) there are far fewer samples" → I would say this is true for all lead times. Except maybe week 1 p = 0.9, although also those are far fewer than p = 0.1.

*Thank you for this remark, however the text has been left as it was, as the forecast week 1 stands out with a larger n also in Figure 5a in the category p>0.66 (in comparison to other weeks' n in Figures 5b-d.*

ll. 356 – 357: could maybe explicitly say that here, you transform the probabilistic forecast to a categorical one. I find this quite appealing by the way, because I think this is a much more likely scenario of how such a forecast would be used in practice. I think it could also be fair to mention that. Figure 5: If you want to, there is a lot more that you could unpack from this figure and the way you look at the forecasts here (i.e., as categorical forecasts). While in this figure, it is easy to see what the outcome was, given a certain forecast (category) was issued, it could be quite insightful to see what the forecast was, given a certain outcome (heat wave/no heat wave). You can actually do this (approximately) with the numbers in the plot. For instance, the forecasts for all lead times have a high likelihood (decreasing with lead time but > 90% for all) $p(f0|o0)$ of forecasting no heat wave when there turned out to be no heat wave, i.e. they have high specificity. However, when you wrap up the ($0.33 < p < 0.66$) and ($p > .66$) categories into one "forecast of heat wave" category, the true positive rate (sensitivity) $p(f1|o1)$ drops from 86% in week 1 to 19% in week 4. In addition, the likelihood $p(f0|o1)$ of forecasting "no heat wave" when there was actually a heat wave, goes up from 18% in week 1 to almost 99% in week 4. This means that if a forecast at longer lead times indicates "heat wave", there are good chances there will be one, but if it shows "no heat wave", you shouldn't rely on there being none. In my eyes, this could be quite important information to someone designing an early-warning system. I leave it up to you, but in my opinion something like this would make a very nice addition to the paper and it could add some points that could be addressed in the discussion.

*Thank you for these remarks, you are making good points. We have edited this to be "Next, we conducted verification of heat wave day forecasts based on forecasted probabilities falling within the ranges of here defined as low: p <0.33, intermediate: 0.33 ≤ p ≤ 0.66, and high: p > 0.66, i.e., we transformed the probabilistic forecast to a categorial one."*

l. 415: "(days 29..35)" → "(days 29 to 35)"?

*Thank you for this remark, we have edited this to be "(days 29 to 35)".*

l. 494: Does ERF stand for "extended-range forecast"? Don't think this is defined anywhere.

*Thank you for this remark, we have added the definition for ERF here.*